# Agentic Framework for Epidemiological Modeling

**Rituparna Datta** [* 1]  **Zihan Guan** [* 1]  **Baltazar Espinoza** [2]  **Yiqi Su** [3]  **Priya Pitre** [3]  **Srini Venkatramanan** [2]
**Naren Ramakrishnan** [3]  **Anil Vullikanti** [1 2]

## Abstract

Epidemic modeling is essential for public health planning, yet traditional approaches rely on fixed model classes that require manual redesign as pathogens, policies, and scenario assumptions evolve. We introduce EPIAGENT, an agentic framework that automatically synthesizes, calibrates, verifies, and refines epidemiological simulators by modeling disease progression as an iterative program synthesis problem. A central design choice is an explicit epidemiological flow graph intermediate representation that links scenario specifications to model structure and enables strong, modular correctness checks before code is generated. Verified flow graphs are then compiled into mechanistic models supporting interpretable parameter learning under physical and epidemiological constraints. Evaluation on epidemiological scenario case studies demonstrates that EPIAGENT captures complex growth dynamics and produces epidemiologically consistent counterfactual projections across varying vaccination and immune escape assumptions. Our results show that the agentic feedback loop prevents degeneration and significantly accelerates convergence toward valid models by mimicking professional expert workflows.

## 1. Introduction

Epidemic modeling is a core tool for public health planning, providing the analytical framework necessary for policymakers to navigate disease dynamics and evaluate intervention strategies, e.g., (Hethcote, 2000; Brauer et al., 2012; Anderson & May, 1991; Marathe & Vullikanti, 2013; Espinoza

et al., 2023). During the COVID-19 pandemic, these models proved indispensable for guiding high-stakes decisions and informing public health policy (Loo et al., 2024). A typical epidemiological model encodes assumptions about disease transmission, immunity, and population stratification and structure. Simulation-based analyses of these models, which are calibrated using available data (usually limited and noisy), are used to study projections and provide insights about different policy scenarios. The COVID-19 Scenario Modeling Hub (SMH) demonstrates the value of multi-model ensembles in providing robust, long-term projections to guide federal and state-level responses (MIDAS Network, 2025; Reich et al., 2022). Since such projections directly inform resource allocation and clinical policy, the integrity, interpretability, and reliability of the modeling process are paramount. Such efforts have expanded since to support seasonal projections across multiple pathogens, including influenza and RSV. Multiple modeling frameworks such as agent-based models (Chen et al., 2024) and metapopulation models (Porebski et al., 2024) have been used to support this effort, ranging across scenarios pertaining to pathogen characteristics (Truelove et al., 2022), vaccination recommendations (Jung et al., 2024; Loo et al., 2025), non-pharmaceutical interventions (Borchering, 2021).

Epidemiological modeling pipelines today rely heavily on manual model design and expert intervention. When policies change, new variants emerge, or immunity assumptions shift, epidemiologists must redesign model structures, revise transition logic, and re-implement simulators—an iterative and time-consuming process that limits scalability and responsiveness. Large Language Models (LLMs) and agentic AI systems have demonstrated significant potential in automating complex scientific and engineering workflows through iterative generation and refinement, e.g., novel symbolic regression strategies to learn complex models from data (Grayeli et al., 2024). Agent-based frameworks have already been deployed in fields such as code synthesis (Zhang et al., 2024; Wang et al., 2024; Hong et al., 2024) and automated research laboratories (Schmidgall et al., 2025). In these systems, structured artifacts——such as simulation codes or experimental protocols——are generated, and corrected through autonomous feedback loops (Lu et al., 2024). However, such methods have not been explored for epidemi-

[1]Department of Computer Science, University of Virginia, Charlottesville, USA [2]Biocomplexity Institute, Charlottesville, USA [3]Department of Computer Science, Virginia Tech, Alexandria, USA. Correspondence to: Anil Vullikanti <vsakumar@virginia.edu>.

*Proceedings of the 43rd International Conference on Machine Learning*, Seoul, South Korea. PMLR 306, 2026. Copyright 2026 by the author(s).

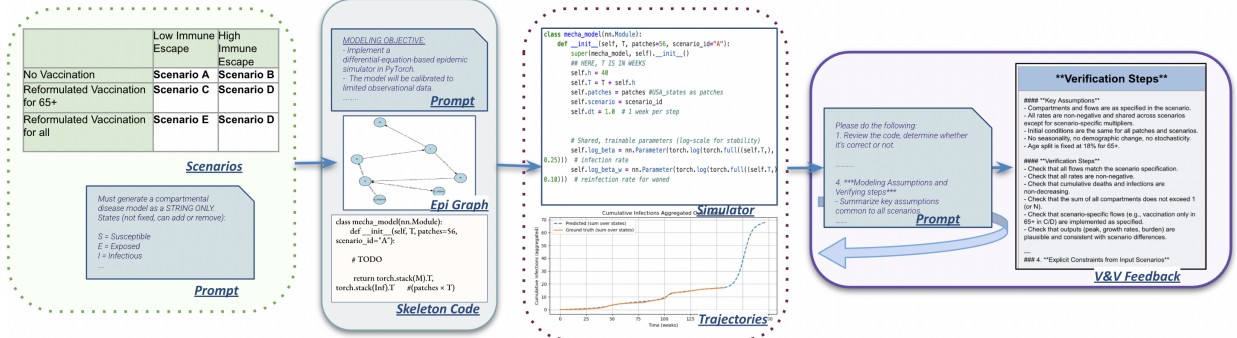

Figure 1. Data flow in EPIAGENT has the following structure: *Prompt* $\leftrightarrow$ *Flow Graph* $\rightarrow$ *Simulator Code* $\rightarrow$ *Results*. The flow graph is an abstract representation of the epidemic model being learned, and supports verification. The flow graph is transformed into the actual epidemic model and is calibrated, and fully verified. These components correspond to the architecture of EPIAGENT in Figure 2.

ological models, a significantly more sophisticated task than the aforementioned science applications. The motivating question for our work is thus: *can agentic systems be developed to assist in epidemiological modeling and analysis?*

We show how naively prompting LLMs to generate compartmental models and simulators leads to frequent structural inconsistencies and fragile behavior. We introduce EPIAGENT, an agentic framework for constructing epidemic models from natural-language epidemic study descriptions; a crucial component in EPIAGENT is a flow graph representation, which helps improve the robustness of the learned models, as illustrated in Figure 1. Our main contributions are: **(1.)** We introduce *retrieval-augmented flow graph synthesis*, which bridges scenario specifications and mathematical structure by generating epidemiological flow graphs that satisfy hard structural constraints, and valid compartmental transitions, and can be verified more easily. **(2.)** We present a *graph-to-model compilation* mechanism that transforms verified flow graphs into mechanistic simulators providing an explicit structural scaffold for disease parameter learning with the real-world data. **(3.)** We design a *multi-agent verification and validation architecture* in which specialized agents enforce correctness properties—mathematical validity, scenario fidelity, and mechanistic interpretability—and provide actionable feedback, mirroring expert epidemiological modeling workflows. **(4.)** We evaluate EPIAGENT on epidemiology modeling studies (MIDAS Network, 2025), and a behavioral SEIR baseline (Gozzi et al., 2025), demonstrating that the generated simulators achieve accurate fits to observed trajectories, produce epidemiologically consistent counterfactual projections, and converge to valid model structures more reliably than unguided generation.

Highlights of EPIAGENT include scaffolded autonomy for robust model generation and flow-graph representations instead of hard-coded constraints (Section C). Future work will improve robustness under incomplete scenario descriptions (Section C.1).

## 2. Related Work

**Epidemiological Modeling.** This is a core tool for public health planning (Marathe & Vullikanti, 2013; Venkatramanan et al., 2019; Espinoza et al., 2020; Mahmud et al., 2025; Dixit et al., 2023), since data is usually limited, and policy analysis questions cannot be solved simply using machine learning methods. Large-scale efforts such as the COVID-19 Scenario Modeling Hub demonstrate the value of ensemble-based evaluation across diverse mechanistic and statistical models (MIDAS Network, 2025; Howerton et al., 2023; Shea et al., 2023). This highlights the challenge of maintaining structural consistency and interpretability under distribution shift, as evolving policies, variants, and immunity assumptions often require manual redesign of model structure rather than parameter re-estimation.

**Discovering equations and physical models from data.** This class of work includes symbolic regression, sparse identification, and system identification aims to recover governing equations from data (Brunton et al., 2016; Rudy et al., 2017; Champion et al., 2019; Ljung et al., 1987; Brunton et al., 2021; Shojaee et al., 2023). Another related topic involves neural ODEs and physics-informed neural networks, which learn continuous-time dynamics with neural components (Chen et al., 2018; Raissi et al., 2019; Karniadakis et al., 2021; Rackauckas et al., 2020; Cuomo et al., 2022). However, epidemic simulators differ in important ways: they encode structured compartment transitions and scenario-driven intervention logic, and are often used for counterfactual and mechanistic reasoning, where black-box components and fixed topologies can hinder interpretability and scientific validity, even when trajectory fits are accurate.

**Agentic Scientific Modeling Systems.** Recent work has explored agentic and automated scientific modeling systems that leverage LLMs for program synthesis, repair, and iterative refinement through execution feedback and tool use (Yao et al., 2022; Madaan et al., 2023; Wang et al., 2024; Zhang et al., 2024; Guan et al., 2026a), as well as

broader 'AI scientist' frameworks for automating research workflows (Lu et al., 2024; Schmidgall et al., 2025) and symbolic regression (Grayeli et al., 2024). These systems primarily target syntactic correctness and empirical performance; typically lack domain-specific mechanistic constraints or support structural revision as a first-class operation. EPIAGENT addresses this gap by treating epidemic simulator construction as an agentic, scenario-conditioned process that explicitly represents and verifies mechanistic structure, integrating verification and validation (V&V) into generation loop to support reliable counterfactual analysis.

## 3. Problem Statement

### 3.1. Preliminaries

A large number of models are used in epidemiology, here, we focus on compartmental epidemic models, (Hethcote, 2000; Brauer et al., 2012; Anderson & May, 1991). A canonical example is the SEIR model, which partitions the population into susceptible $S(t)$, exposed $E(t)$, infectious $I(t)$, and recovered $R(t)$:

$$\frac{dS}{dt} = -\beta\frac{SI}{N}, \ \frac{dE}{dt} = \beta\frac{SI}{N} - \sigma E, \ \frac{dI}{dt} = \sigma E - \gamma I, \ \frac{dR}{dt} = \gamma I$$

where $N = S + E + I + R$ is the total population, $\beta$ is the transmission rate, $\sigma$ is the incubation rate, and $\gamma$ is the recovery rate. Most epidemic analyses involve introducing additional states and transitions (e.g., to represent interventions, immunity, and population heterogeneity). We refer to a specific epidemiological study as a *scenario*, denoted by $s$. This could include (i) assumptions about disease-dynamic (e.g., variants, hospitalization), (ii) interventions (e.g., timing, intensity, and compliance of non-pharmaceutical policies), and (iii) population heterogeneity (e.g., spatial structure, demographic variation, or contact patterns). Let $\mathcal{X}$ denote a partitioning of the population into sub-populations (e.g., based on age groups), referred to as patches. Let $V$ denote the set of health states being represented over all patches in $\mathcal{X}$, and let $\mathbf{x}(t) \in \mathbb{R}^K$ denote the vector of epidemic state values (e.g., fraction of susceptible and infectious people), where $K = |V|$. An epidemic model for scenario $s$, denoted by $f_\theta^s(\mathbf{x}(t))$, provides the state vector values at time $t + 1$. $\theta$ encodes disease and intervention parameters–may include parameters such as $(\beta, \sigma, \gamma)$, or time-varying parameters such as $\beta(t)$ or $\gamma(t)$ to capture behavioral or seasonal effects, or structured parameter functions driven by exogenous covariates. Some studies, such as the CDC Scenario Modeling Hub (MIDAS Network, 2025), require developing models for a *set of scenarios*, denoted $\mathcal{S}$. There is significant theory that can be leveraged about validity of mechanistic models, e.g., non-negativity of parameters, and monotonicity of cumulative quantities, e.g., (Brauer et al., 2012). A simulation for a model $f_\theta^s(\cdot)$ refers to the actual implementation of the dynamical system model.

To better understand model fit and expressive capacity, we also consider *hybrid composition models*. In these models, the parameter set $\theta$ is learned using a neural network (Chopra et al., 2023; Guan et al., 2026b; Datta et al., 2025). Such hybrid models increase flexibility when fitting complex or multi-peak epidemic trajectories, but are hard to interpret due to the black box component. Here, we include such models and study the improvement in fit by neural augmentation, but this is not the main focus of our work.

**Scenario Flow Graph.** Each scenario $s \in \mathcal{S}$ can be represented by a directed epidemiological flow graph, $G = (V, E)$, on the nodes in $V$, where each edge $(u, v) \in E$ denotes an admissible transition, and is parameterized by shared disease parameters $\theta$. The SEIR model, incorporating disease-induced mortality can be represented by the directed flow graph $S \xrightarrow{\beta} E \xrightarrow{\sigma} I \xrightarrow{\gamma} R, \qquad I \xrightarrow{\mu} D$

where $S$, $E$, $I$, $R$, and $D$ denote the susceptible, exposed, infectious, recovered, and deceased compartments, respectively. The parameters $\beta$, $\sigma$, $\gamma$, and $\mu$ correspond to the transmission, progression, recovery, and disease-induced death rates (Brauer et al., 2012).

### 3.2. Problem Definition

Let $\mathcal{D} = \left\{ x_i^{(s)}(p, t), i \in V' \right\}_{s=1, t=1}^{\mathcal{S}, T}$ denote the dataset of observed epidemiological time series data for a subset of node states $V'$ for each patch $p \in \mathcal{X}$ over discrete weekly time steps $t = 1, \ldots, T$. We restrict the model parameters and the compartmental flows to a feasible set of constraints $\mathcal{C} \subseteq \Theta$, defined as

$$\mathcal{C} = \left\{ \theta \in \Theta \mid \mathbf{g}(\theta) \leq \mathbf{0}, \ \mathbf{h}(\theta) = \mathbf{0} \right\}, \tag{1}$$

where $\mathbf{g}(\theta)$ encodes inequality constraints such as disease parameter positivity, boundedness, and numerical stability conditions, and $\mathbf{h}(\theta)$ encodes constraints such as conservation laws and normalization constraints—incorporated in $\mathcal{C}$. Given $\mathcal{D}$ and $\mathcal{S}$, learning an epidemic model corresponds to solving the following optimization problem:

$$\theta^\star = \arg\min_{\theta \in \mathcal{C}} \sum_{p=1}^{P} \sum_{t=1}^{T} \mathcal{L}\left( f_\theta^{\mathcal{S}}(\cdot), \ \left(x_i^{(s)}(p, t)\right) \right) \tag{2}$$

where $\mathcal{L}(\cdot)$ denotes loss function that measures the discrepancy between simulated and observed trajectories.

Given a set of scenarios $\mathcal{S}$, specified using natural language description and observational data $\mathcal{D}$, the goal of EPIAGENT is to produce epidemiological models $f_\theta^s, s \in \mathcal{S}$, and the simulations, satisfying the following properties: (1) *Satisfactory accuracy*: the generated models should minimize the empirical loss as formalized by the optimization objective in Eq (2). (2) *Model validation and verification*: the learned models $f_\theta^s, s \in \mathcal{S}$ should satisfy constraints from epidemiology theory (Eq 1, details in § 4.5). and (3) *Scenario*

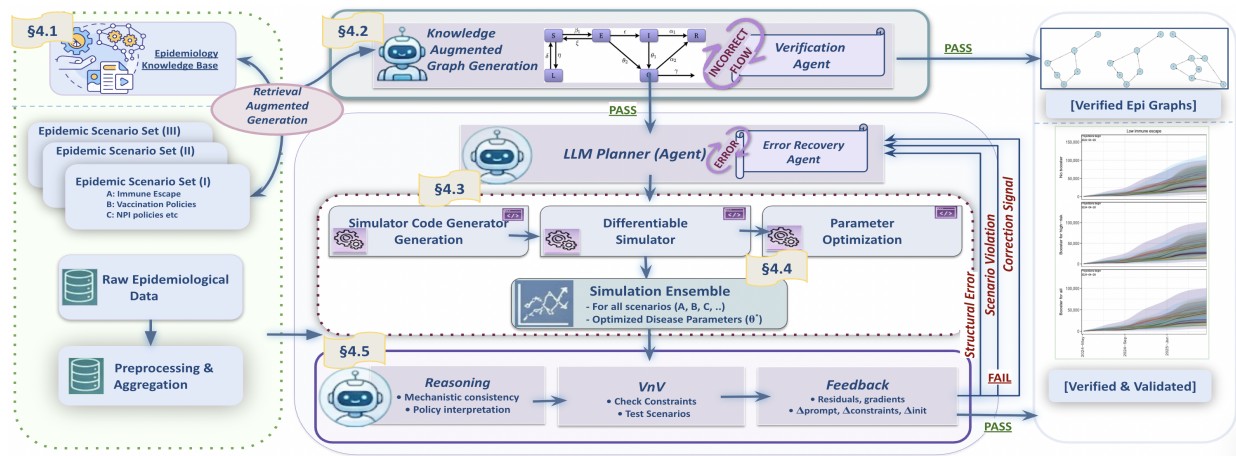

*Figure 2.* Agentic pipeline for epidemic scenario modeling and ensemble simulation. Natural-language scenarios are augmented with domain knowledge (§4.1), to produce prompts that guide flow-graph synthesis (§4.2). Generated graphs are iteratively verified to enforce valid compartmental structure and transitions. Given a verified graph and scenario description, an LLM planner instantiates executable simulator code with automated error recovery (§4.3). Simulators are calibrated on observed data and evaluated as a scenario ensemble (§4.4). A multi-agent verification and validation stage enforces epidemiological and scenario-consistency constraints, retaining only structurally and behaviorally valid models (§4.5).

*consistency and validity*: the learned models should remain internally consistent across scenarios and faithfully reflect each scenario's assumptions, interventions, and constraints.

## 4. Method

We introduce EPIAGENT, an agentic framework for epidemiological modeling. Figure 1 presents a high-level view of the system's data flow from natural-language scenario prompts to simulation outputs, while Figure 2 details the internal agentic pipeline that implements this architecture. Below, we describe the main components of EPIAGENT.

### 4.1. Scenario Parsing and Knowledge Retrieval

The first phase of EPIAGENT takes a natural-language scenario specification $\mathcal{S}$ as input and outputs a knowledge-augmented user-generated prompt encoding explicit structural and scenario constraints $\mathcal{C}$ for flow-graph generation.

For extracting relevant knowledge, epidemic descriptions are embedded using a sentence-transformer model and augmented with relevant epidemiological knowledge retrieved from a curated corpus using a FAISS-based vector index (Reimers & Gurevych, 2019; Douze et al., 2024). Retrieved passages are combined with the specification to form a knowledge-augmented prompt, grounding model construction in established epidemiological theory (Gao et al., 2023).

### 4.2. Knowledge-Augmented Synthesis of Epidemiological Flow Graphs

The epidemiological flow graph $\mathcal{G} = (V, E)$ forms the structural basis of the simulator. We generate it through

an iterative, knowledge-conditioned process guided by epidemiological constraints and scenario assumptions, with a verification loop enforcing structural correctness of compartments and transitions.

Graphs that violate these constraints are rejected and regenerated using explicit error feedback. Graphs that pass these checks are then subjected to an *agentic verification* loop that evaluates scenario consistency and requires epidemiological justification of the proposed transitions. During this step, structured feedback is injected to correct specific deficiencies (e.g., missing waning transitions, invalid vaccination flows, or inconsistent compartment semantics), producing a revised graph. Verification is automated by (i) checking alignment between compartments and scenario assumptions (e.g., vaccination eligibility, immune escape, and waning), and (ii) enforcing explicit epidemiological justification for each transition. The loop terminates only when the graph is verified as valid under both structural and scenario-specific criteria. If verified as valid, the loop terminates. This graph-level verification prevents invalid structures from propagating into executable code. A sample graph-verification response is shown in Appendix Figure 16.

### 4.3. Dynamical System Model and Simulator Generation

Given a verified flow graph and disease specifications, EPIAGENT instantiates a dynamical model and its executable simulator, $f_\theta^s(\cdot)$. The graph defines the system of ordinary differential equations, after which simulator code is automatically generated and calibrated against ground truth.

**Skeleton Code and Execution Interface.** To ensure exe-

cutability and reproducibility, we provide the model with a code skeleton, $\mathcal{CK}$, that fixes the execution interface while leaving epidemiological logic unconstrained. It defines environment-dependent components—module structure, time indexing, and scenario identifiers—and requires the simulator to implement a differentiable forward pass outputting scenario-specific target trajectories (e.g., infection and death counts) (Fig 13, Appendix). Separating the fixed interface from the model logic enables automated execution, training, and validation without manual intervention.

**Constraint-Guided Prompting.** Alongside the skeleton code, we impose a set of hard constraints $\mathcal{C}$ that guide generation and prevent common failure modes. Derived from external epidemiological knowledge or hardcoded, they fall into two categories: (i) constraints enforcing valid epidemic structure, such as numerical stability and compartmental consistency, and (ii) execution constraints preventing unsafe coding behaviors, such as reassigning trainable parameters within the forward pass. We deliberately avoid enforcing parameter non-negativity through ad hoc clipping; instead, violations indicate that the model itself must be revised. Epidemiological principles are provided as soft constraints to encourage interpretable designs without limiting expressiveness. Together, these constraints ensure generated simulators are both executable and scientifically meaningful.

**LLM Planner Agent, $\mathcal{A}_{plan}$.** This planning agent generates differentiable simulator code by instantiating predefined execution skeletons with compartmental structures derived from the verified flow graph, conditioned on the given specifications. It has an *Error Recovery Module* that monitors execution during compilation, simulation, and training. When runtime, numerical, or structural errors occur (e.g., shape mismatches, invalid tensor operations, solver instability, or differentiability violations), it captures the full error trace and returns it as structured feedback to the coding agent, which revises the implementation while preserving the prescribed skeleton and scenario constraints. This loop runs for a bounded number of retries, ensuring robust autonomous code generation without manual intervention.

### 4.4. Training and Calibration

For each generated simulator, disease parameters $\theta$ are calibrated to observational data using gradient-based optimization. We use a mean squared error (MSE) loss over infection and deceased levels and first-order differences:
$\mathcal{L} = \frac{1}{P} \sum_{p=1}^{P} \left( \text{MSE}(\hat{x}_p, x_p) + \text{MSE}(\Delta \hat{x}_p, \Delta x_p) \right)$

Where, $\Delta x_t = x_{t+1} - x_t$. Parameters are shared across scenarios unless explicitly overridden by the scenario specification, ensuring that scenario-dependent differences arise from modeled mechanisms rather than independent re-fitting. The calibrated simulator is then evaluated by generating projections across all scenarios. Optimization hyperparameters,

including the optimizer and learning-rate schedule, are fixed and predefined; the agent is restricted to completing the simulator skeleton and cannot modify the training environment.

### 4.5. Multi-Agent Verification and Validation (V&V)

V&V is enforced through automated LLM-driven modules.

**Verification Module.** This module checks the structural and numerical correctness of the simulator, including: (i) consistency between the executable code $f_\theta^s(\cdot)$ and the verified flow graph $G$, (ii) non-negativity of states and parameters: For all compartments $x_i(t) \geq 0$ for all $t$, preventing unphysical population counts, (iii) conservation of population mass: The system must satisfy $\sum_i \frac{dx_i}{dt} = 0$, ensuring the total population remains constant, *excluding deceased flows*, and (iv) numerical stability during simulation.

**Model Reasoning Module.** This module analyzes and explains the generated code and model structure, providing explicit justifications for compartment choices, parameterization, and dynamical assumptions, thereby improving interpretability and transparency.

**Validation Module**. This module evaluates plausibility and scenario fidelity, assessing whether simulated outcomes reflect intended scenario mechanisms and cross-scenario differences across all evaluated scenarios.

Finally, the **Performance Feedback Module** evaluates calibration and validation results (e.g., loss trends, generalization gaps) and produces targeted feedback to guide iterative model refinement toward improved predictive performance. The full procedure is formalized in Algorithm 1, Appendix.

## 5. Experimental Setup

**Datasets.** We evaluate our framework using scenario modeling case studies from the COVID-19 Scenario Modeling Hub archive, which comprises 19 scenario rounds (*Case Study I*). Each round specifies structured assumptions over interventions, vaccination, and immunity, together with corresponding epidemiological time series for projection and evaluation (MIDAS Network, 2025). We additionally consider *Case Study II*, which evaluates calibration and probabilistic projection on real-world epidemiological data using a standard age-structured behavioral SEIR model. This study uses publicly available COVID-19 surveillance data from four heterogeneous locations, including reported deaths, mobility-derived behavioral signals, and seasonality covariates (Gozzi et al., 2025). Compartmental structures and transition semantics are derived from established formulations in mathematical epidemiology (Brauer et al., 2012), which serve as the epidemiological knowledge base queried via retrieval-augmented generation (RAG).

**LLM Selection.** Epidemiological flow graphs are generated

using `gpt-4o-mini`(Achiam et al., 2023). Knowledge retrieval is implemented using sentence-transformer embeddings (`all-MiniLM-L6-v2`) indexed with FAISS, which enables efficient semantic retrieval from epidemiology textbooks. We use `gpt-4.1` for planning and simulator code generation, as these stages require higher reasoning capacity, while verification, validation, and feedback agents use the lighter-weight `gpt-4.1-mini` model to reduce computational cost without sacrificing correctness. We also evaluate open-source (Qwen-Coder) and frontier (GPT-5.2, Opus 4.6) models as planner/generator, finding that simpler models suffice for EPIAGENT (§D.2).

**Agentic Configuration.** LLM agents operate under controlled and adaptive sampling. The temperature is initialized at zero to encourage deterministic generation and is increased only when repeated or cyclic responses are detected. Token budgets, rate limits, and a fixed maximum number of retries are enforced to bound inference cost.

**Execution Environment.** All simulators are calibrated by completing a fixed PyTorch execution environment and are trained in a predefined setting with a fixed optimizer (Adam) and learning-rate scheduler (ReduceLROnPlateau).

**Hardware Configuration.** All experiments are conducted using PyTorch with GPU acceleration on NVIDIA Tesla V100-SXM2 GPUs (32GB memory) with CUDA.

## 5.1. Ablation Study

To isolate the contribution of EPIAGENT's key design choices, we conduct targeted ablations that remove (i) flow-graph verification, (ii) access to external epidemiological knowledge, and (iii) the flow-graph intermediate representation (i.e., direct scenario-to-code generation). We evaluate each variant by inspecting the validity of the induced model structure, the frequency and severity of structural errors, and the number of generate–verify iterations required to reach a mechanistically valid simulator.

**Without Flow-Graph Verification.** Omitting graph-level verification results in epidemiologically invalid structures that propagate into dynamical models. Across our experiments, incorrect flow graphs result in incorrect scenario projections in over 87% of cases, underscoring the necessity of explicit graph-level verification.

**Without External Knowledge Retrieval (No RAG).** Without epidemiological knowledge retrieval, the LLM defaults to the simplest SEIRD-style graphs, often omitting scenario-specific mechanisms such as waning immunity, or age-targeted vaccination. Empirically, knowledge guidance reduces the average number of graph-generation iterations with feedback by $\sim 20\%$. (Figure 9 and 10, Appendix).

**Without Flow-Graph Intermediate Representation (Di-**

**rect Scenario-to-Code).** Direct equation generation from scenario text often produces structural errors, such as missing compartments, duplicated states, or invalid transitions (e.g., $R \to D$) (Figure 11, Appendix). These errors are difficult to detect and recover at the code level. Introducing an explicit flow-graph intermediate representation decouples structural validation from equation synthesis, enabling targeted verification and substantially improving reliability.

## 6. Results

We evaluate the proposed agentic framework through two complementary case studies to assess calibration accuracy, structural correctness, and counterfactual reasoning. *Case Study I* evaluates whether agent-generated simulators correctly interpret and execute complex counterfactual scenarios. *Case Study II* evaluates whether EPIAGENT can recover and deploy a standard behavioral epidemiological model for fitting and probabilistic projection across real-world data. Together, these case studies evaluate EPIAGENT by addressing the following questions.

1. Can EPIAGENT calibrate epidemic dynamics? (§6.1)
2. Does graph verification avert incorrect solutions? (§6.2)
3. Does code-level verification prevent degenerate low-loss solutions (e.g., negative states, mass imbalance)? (§6.2)
4. Do scenario-conditioned models respond correctly to counterfactual interventions? (§ 6.3)
5. Does agentic feedback improve performance? (§6.4)
6. Can EPIAGENT be applied to classical epidemiological models to generate reliable projections? (§6.5)

*Table 1.* Performance comparison of modeling approaches.

| | Model | MAE | MSE | RMSE |
|---|---|---|---|---|
| **State-avg** | Time-invariant dynamical | $3.84 \times 10^{-3}$ | $3.10 \times 10^{-5}$ | $5.50 \times 10^{-3}$ |
| | Time-variant $(\beta_t, \gamma_t)$ | $8.90 \times 10^{-4}$ | $1.30 \times 10^{-6}$ | $1.10 \times 10^{-3}$ |
| | Neural-incorporated | $6.27 \times 10^{-4}$ | $7.50 \times 10^{-7}$ | $8.00 \times 10^{-4}$ |
| **Nationwide** | Time-invariant dynamical | $2.12 \times 10^{-1}$ | $9.57 \times 10^{-2}$ | $3.09 \times 10^{-1}$ |
| | Time-variant $(\beta_t, \gamma_t)$ | $4.26 \times 10^{-2}$ | $2.53 \times 10^{-3}$ | $5.03 \times 10^{-2}$ |
| | Neural-incorporated | $2.22 \times 10^{-2}$ | $7.13 \times 10^{-4}$ | $2.67 \times 10^{-2}$ |

While EPIAGENT is designed for general epidemiological modeling, we also use a specific COVID-19 scenario-modeling round as a primary case study to demonstrate its counterfactual reasoning capabilities. For comparison, we also evaluate general-purpose LLM agents (Nathani et al., 2025; Holt et al., 2024) and zero-shot prompting of frontier models–GPT 5.2 and Claude Opus 4.6 (§D.1).

## 6.1. Empirical Calibration of Generated Simulators

We first evaluate the empirical calibration of the simulators produced by EPIAGENT by comparing their projected trajectories against observed epidemiological data. Figure 3 shows the fit of the simulator with time variant disease parameters with uncertainty. Allowing time-varying disease parameters (e.g., $\beta(t), \gamma(t)$) improves short-term forecast-

ing accuracy (Table 1). When additional flexibility is required, the agent can introduce neural components to learn time-varying parameters. By default, however, the agent restricts the model to time-invariant parameters, avoiding unnecessary over-parameterization while preserving interpretability. *Hybrid formulations* are particularly well-suited for forecasting settings, where maximizing predictive accuracy is prioritized. Despite this, the generated models are able to capture key temporal characteristics, including overall growth dynamics and peak timing, which are the primary objectives of scenario-based projection.

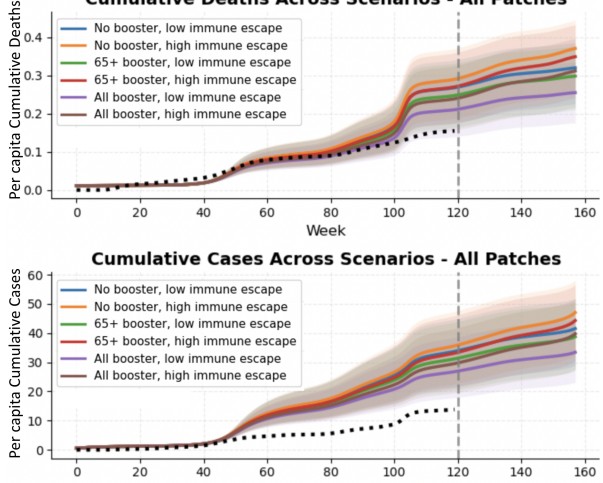

*Figure 3.* National-level cumulative COVID-19 infections across six scenarios from (MIDAS Network, 2025). State-level trajectories are population-normalized and aggregated into nationwide infection and death curves. The dashed vertical line marks the boundary between training and projection periods, while the black dotted line denotes observed data.

## 6.2. Model Verification

We evaluate the effectiveness of the proposed verification mechanism in enforcing epidemiological and structural correctness of the synthesized simulators. Verification operates at both the *structural (flow-graph)* and *behavioral (code execution)* levels, and serves as a filter that eliminates epidemiologically invalid models.

**Mechanistic validity.** Models generated without verification frequently exhibit (i) invalid state transitions (e.g., direct $S-> R$ recovery without infection), (ii) implausible causal pathways (e.g., deaths arising from susceptible), and (iii) missing required flows (e.g., lack of waning immunity transitions under scenarios specifying immune escape). Using retrieval-augmented domain knowledge, the system identifies and rejects such models at the flow-graph level, ensuring that only graphs consistent with established epidemiological principles are admitted for simulator generation.

*To enforce mechanistic validity,* EPIAGENT *explicitly sepa-*

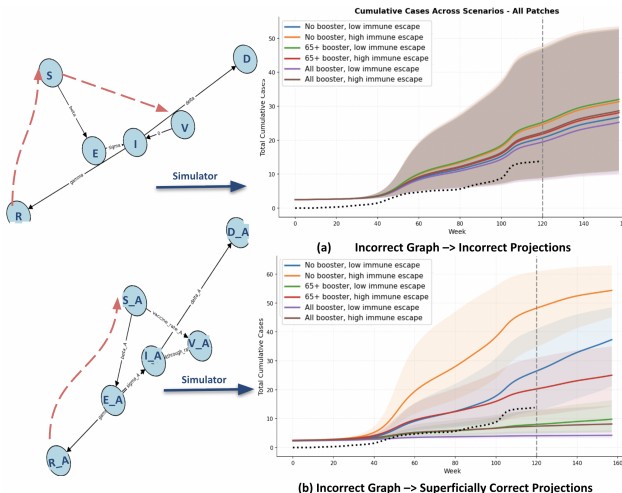

*Figure 4.* Error propagation from incorrect graphs to epidemic projections: may yield plausible outcomes (b) or incorrect trajectories (a)–demonstrating the necessity of structural verification.

*rates natural-language prompting from simulator synthesis by first generating and verifying an intermediate compartmental flow graph.*

**Impact of Graph Correctness Agent.** In EPIAGENT, the objective is scenario modeling rather than point forecasting; consequently, correctness of the compartment graph is essential for causal validity and counterfactual reliability. While an incorrect graph often leads to incorrect projections, this is not the only failure mode. In some cases, parameter flexibility can compensate for structural mis-specification, producing epidemic curves that appear numerically reasonable despite encoding incorrect causal mechanisms.

We categorize graph–projection outcomes into three cases: (i) *correct graph, correct result (Figure 3)*; (ii) *incorrect graph, incorrect result*; and (iii) *incorrect graph, correct-looking result*. Figure 4 illustrates the latter two cases.

In Figure 4(a), the compartment graph omits population-level immunity loss (no $R \rightarrow S$ or waning state), which structurally constrains the dynamics by suppressing reinfection. As a result, scenario projections are biased and diverge for incorrect mechanistic reasons, yielding visibly incorrect cumulative trajectories. In contrast, Figure 4 (b) also relies on an incorrect graph, but sometimes, flexible parameterization partially masks the structural error, yielding aggregate trajectories that appear consistent across scenarios.

**Parameter and state constraints.** Verification eliminates models that violate fundamental epidemic constraints during execution. Without enforcement, gradient-based calibration can converge to numerically low-loss but epidemiologically invalid solutions. Common failure modes include: (i) negative compartment values (e.g., $I(t) < 0$ or $E(t) < 0$), which are physically meaningless; (ii) violation of population conservation, where $S(t) + E(t) + I(t) + R(t)$ exceeds or falls

below the population size due to inconsistent flows; and (iii) non-monotonic cumulative quantities, such as cumulative deaths decreasing over time after calibration. The verification agent enforces non-negativity, population conservation, and monotonicity, preventing such degenerate but low-loss solutions from being accepted (Figure 14, Appendix).

### 6.3. Scenario Validation

We validate the synthesized simulators through counterfactual analysis across scenarios. By holding the calibrated parameterization fixed and varying only the scenario specification, we examine whether projected outcomes respond in a coherent and epidemiologically interpretable manner. Differences in vaccination strategy and immune escape assumptions produce systematic shifts in projected infections, hospitalizations, and mortality, demonstrating that the models not only fit historical data but also support meaningful scenario-driven reasoning. We evaluate this capability using a structured scenario in *Case Study I* (Fig 8, Appendix).

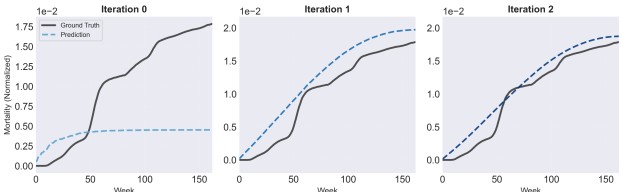

*Figure 5.* Iterative refinement of epi models under agentic verification and feedback. Subsequent feedback corrects structural violations and progressively refines model expressiveness.

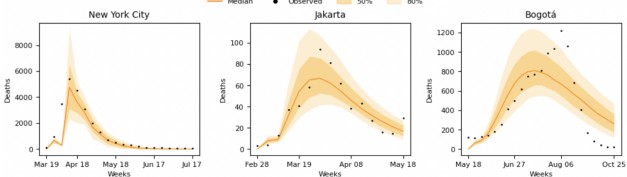

*Figure 6.* Fitted curves of weekly COVID-19 deaths (counts) across the three geographies considered, using the Data-Driven Behavioral epidemiological model instantiated by EPIAGENT.

**Case Study I.** We evaluate six scenarios (A–F) with two variations: *vaccination policy*—(i) no recommendation with negligible uptake (A, B), (ii) targeted boosters for high-risk populations aged 65+ (C, D), and (iii) universal booster for all eligible groups (E, F)—and *immune escape level* (low vs. high). As shown in Figure 3, the synthesized simulators exhibit strong epidemiological consistency. Importantly, these qualitative behaviors are *not manually encoded*, but are inferred by the agent through scenario-conditioned graph construction and simulator synthesis. Within each policy pair, high immune escape scenarios (B, D, F) lead to higher cumulative infections than low-escape cases (A, C, E). Increasing vaccination coverage consistently reduces infections ($E < C < A$ and $F < D < B$), reflecting

correct counterfactual behavior in the learned model.

### 6.4. Effect of Agentic Feedback on Convergence

We find that iterative agentic feedback significantly accelerates convergence compared to unguided generation. As illustrated in Figure 5, subsequent feedback cycles correct structural violations and refine model expressiveness—such as adding waning-immunity pathways or vaccination strata—thereby mimicking a professional expert's workflow (Figure 15, Appendix).

### 6.5. Behavioral Epidemic Models Synthesis

To evaluate the generality of EPIAGENT, we conducted a baseline case study, (*Case Study II*) following the Data-Driven Behavioral (DDB) epidemiological model introduced in (Gozzi et al., 2025). A DDB model incorporates behavioral changes by leveraging mobility data to capture variations in contact patterns. Using EPIAGENT, we instantiated the DDB model and applied it across diverse locations. For each location, the DDB model generated by EPIAGENT is calibrated to weekly aggregated deaths using reported surveillance data. Probabilistic projections are then obtained by constructing a parametric ensemble around the calibrated parameters. Figure 6 reports the median projected trajectory together with 50% and 80% uncertainty intervals, overlaid with observed weekly deaths. Across all three locations, the projections capture key epidemic trends and peak timing, which demonstrates that EPIAGENT can be directly applied to any canonical epidemiological models to produce calibrated, uncertainty-aware projections. Importantly, this case study shows that EPIAGENT can serve as a general framework for deploying, calibrating, and analyzing standard behavioral epidemic models.

## 7. Conclusion

We present EPIAGENT, an end-to-end agentic framework designed to automate the scenario-conditioned epidemic modeling. Our multi-agent verification and validation architecture successfully bridges the gap between abstract public health scenarios and executable, mechanistically sound simulators. Our evaluation through case studies confirms that the framework not only achieves high empirical accuracy but also maintains structural integrity and logical consistency across complex counterfactual interventions. While effective, the current system relies on fixed optimization and bounded iterative refinement, which may occasionally lead to late-stage regression due to compounding noise. Future work will investigate convergence-aware calibration and adaptive stopping criteria to improve robustness. Overall, EPIAGENT demonstrates the potential of agentic systems to emulate expert epidemiological workflows and accelerate reliable decision support in evolving public health settings.

## Impact Statement

This work advances machine learning for scenario-conditioned epidemiological modeling to support public health analysis. The framework is intended strictly as a decision support and modeling tool, improving scalability and interpretability while enforcing epidemiological validity. We do not anticipate significant ethical or societal risks; our framework does not generate or design pathogens, or provide guidance for laboratory experimentation.

## Acknowledgement

This work was supported by the MIDAS Coordination Center (NIGMS grant R24GM153920), US National Science Foundation grants DBI-2412389, CCF-1918770, IIS-2312794, CCF-1918656, CNS-2317193, and by Cooperative Agreement CDC-RFA-FT-23-0069 from the CDC's Center for Forecasting and Outbreak Analytics. Any opinions, findings, and conclusions or recommendations expressed in this material are those of the author(s) and do not necessarily reflect the views of the sponsor(s).

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

## A. Overview of EPIAGENT

Algorithm 1 summarizes the end-to-end workflow of EPI-AGENT, specifying inputs, outputs, and the sequence of agentic generation steps, while Algorithm 2 focuses on the verification and validation of the generated epidemiological models, which are crucial components of our framework.

---

**Algorithm 1** Agentic Epidemiological Model Generation

---

**Require:** Scenario $\mathcal{S}$, $\mathcal{D}$, constraints $\mathcal{C}$, code skeleton $\mathcal{CK}$
**Ensure:** Validated, scenario-consistent simulator, $f_\theta^\mathcal{S}(\cdot)$, Optimized parameters $\theta^*$, Reasoning $\mathcal{R}$
1: {**Stage I: Knowledge-Augmented Graph Synthesis**}
2: $\mathcal{K} \leftarrow \text{RAG}(\{\mathcal{S}\}, \text{EpiKnowledgeBase})$
3: $\mathcal{P}_{graph} \leftarrow \text{PROMPT}(\{s_i\}, \mathcal{C}, \mathcal{K})$
4: **repeat**
5:     Generate flow graph $\mathcal{G}_g \leftarrow \mathcal{A}_{graph}(\mathcal{P}_{graph})$
6:     $\mathcal{V}_{graph} \leftarrow \text{GRAPH\_VERIFICATION}(\mathcal{G}_g, \mathcal{C})$
    {Check transitions & Agentic Verification}
7:     $\mathcal{P}_{graph} \leftarrow \mathcal{P}_{graph} \oplus \mathcal{A}_{Feedback}(\mathcal{V}_{graph})$
8: **until** $\mathcal{V}_{graph} = \text{PASS}$
9: {**Stage II: Iterative Functional Implementation**}
10: $\mathcal{P}_{code} \leftarrow \text{PROMPT}(\mathcal{G}_g, \mathcal{CK}, \mathcal{S})$
11: **for** generation $g = 1, \ldots, G$ **do**
12:     Mechanistic Simulator $\mathcal{C}_g \leftarrow \mathcal{A}_{planner}(\mathcal{P}_{code}, \mathcal{G}_g)$
13:     **if** $\mathcal{C}_g$ execution fails **then**
14:         $\mathcal{P}_{code} \leftarrow \mathcal{P}_{code} \oplus \text{ERRORTRACE}(\mathcal{C}_g)$
15:         **continue**
16:     **end if**
17:     {**Stage III: Hybrid Calibration & V&V**}
18:     **Train:** $\theta_g \leftarrow \arg\min_\theta \mathcal{L}(\mathcal{C}_g, \mathcal{D})$
19:     $\mathcal{R}_g \leftarrow \text{GET\_STRUCTURALREASONING}(\mathcal{C}_g, \theta_g)$
20:     $\mathcal{V}_g \leftarrow \text{CHECK\_V\&V}(\mathcal{C}_g, \theta_g, \{\mathcal{S}_i\})$
21:     **if** $\mathcal{V}_g = \text{PASS}$ and $\mathcal{L} < \epsilon$ **then**
22:         **return** $(f_{\theta_g}, \theta_g, \mathcal{R}_g)$
23:     **else**
24:         $\mathcal{F}_g \leftarrow \text{GET\_FEEDBACK}(\mathcal{V}_g, \mathcal{L})$
25:         $\mathcal{P}_{code} \leftarrow \mathcal{P}_{code} \oplus \mathcal{F}_g$
26:     **end if**
27: **end for**
    *Return* Final validated simulator code $\mathcal{C}^*$ and optimized parameters $\theta^*$

---

Table 2 lists the notation used throughout the paper.

## B. Extended Ablation Study

**Without Flow-Graph Verification** When the verification step is removed, the LLM frequently produces epidemiologically invalid transitions, such as direct $V->E$ flows, which violate basic disease progression semantics. Incorporating a verification layer that enforces structural constraints (e.g., valid infection pathways and intervention eligibility)

```
| Scenario | Should differ by    | What to check in output   | What is in the output       | Is consistent? |
|----------|---------------------|---------------------------|-----------------------------|----------------|
| A        | No booster, low escape  | High burden, moderate peak | High infections/deaths      | Yes            |
| B        | No booster, high escape | Highest burden, highest peak | Highest infections/deaths | Yes            |
| C        | 65+ booster, low escape | Lower deaths than A, lower peak | Lower than A, higher than E | Yes        |
| D        | 65+ booster, high escape| Lower than B, higher than C | Lower than B, higher than C | Yes           |
| E        | All booster, low escape | Lowest burden, lowest peak | Lowest infections/deaths    | Yes            |
| F        | All booster, high escape| Lower than B/D, higher than E | Lower than B/D, higher than E | Yes        |
```

*Figure 7.* Validation Agent generated scenario-driven differences and validation checks.

|  | **Low immune escape**
• Immune escape occurs at a constant rate of **20% per year** | **High immune escape**
• Immune escape occurs at a constant rate of **50% per year** |
|---|---|---|
| **No vaccine recommendation**
• Uptake negligible or continues at very slow levels based on existing 2022 booster trends | Scenario A | Scenario B |
| **Reformulated annual vaccination recommended for 65+ and immunocompromised**
• Reformulated vaccine has **65% VE against variants circulating on June 15**
• Vaccine becomes **available September 1**
• Uptake in 65+ same as first booster dose recommended in September 2021
• Uptake in individuals under 65 negligible or continues to trickle based on 2022 booster trends | Scenario C | Scenario D |
| **Reformulated annual vaccination recommended for all currently eligible groups**
• Reformulated vaccine has **65% VE against variants circulating on June 15**
• Vaccine becomes **available September 1**
• 65+ uptake same as first booster dose recommended in September 2021
• Coverage in individuals under 65+ saturates at levels of the 2021 booster (approximately 34% nationally) | Scenario E | Scenario F |

*Figure 8.* Covid19 Scenario modeling: round 17

is therefore necessary to ensure the correctness of the generated flow graphs. Figure 4 illustrates how errors in graphs may propagate and lead to incorrect solutions.

**Without External Knowledge Retrieval (No RAG)** Removing external epidemiological knowledge retrieval results in simpler flow structures, typically defaulting to canonical SEIRD-style models without explicit quarantine or isolation compartments. In contrast, when domain knowledge is retrieved and provided to the LLM, the generated graphs consistently include epidemiologically meaningful compartments and transitions (e.g., exposed, quarantined, isolated), aligned with the given scenarios. An additional observation is that with knowledge augmentation, the generated graphs often satisfy structural constraints in the first iteration, whereas without retrieval, multiple generate–verify cycles are required for the LLM to converge to a valid flow.

**Without Flow-Graph Intermediate Representation (Direct Scenario-to-Code)** In this ablation, the LLM is prompted to generate executable model equations directly from the scenario description, bypassing the intermediate flow-graph representation. We observe that this approach leads to higher rates of structural errors in the generated equations, including missing compartments and invalid transitions (i.e., $R- > D$), which are harder to detect from the code. Introducing an explicit flow-graph intermediate representation improves reliability by decoupling structural validation from equation generation and enabling targeted verification before code synthesis.

## C. Novelty and Design Rationale

### Main Novelty

1. Standard pipelines are insufficient. As we discuss in the paper, we find that the standard agentic code generation and verification pipeline does not work well. A model may fit observed trajectories well while still encoding an epidemiologically incorrect structure, so

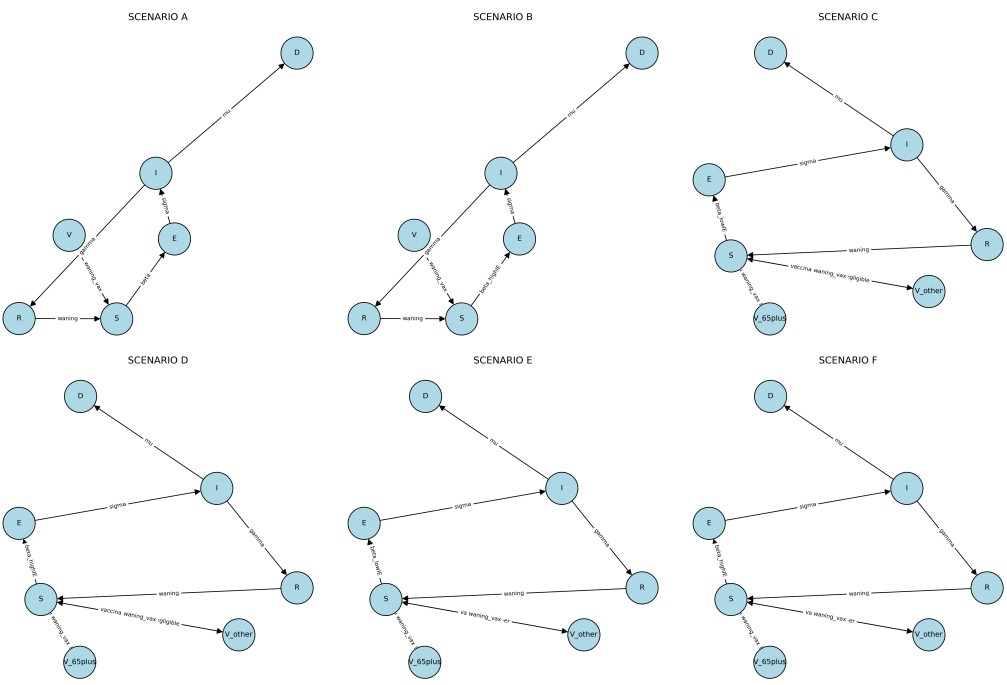

*Figure 9.* Scenario-based epidemiological flow graph construction. *Retrieved domain knowledge constrains graph structure*, enabling the LLM planner to generate compartmental models.

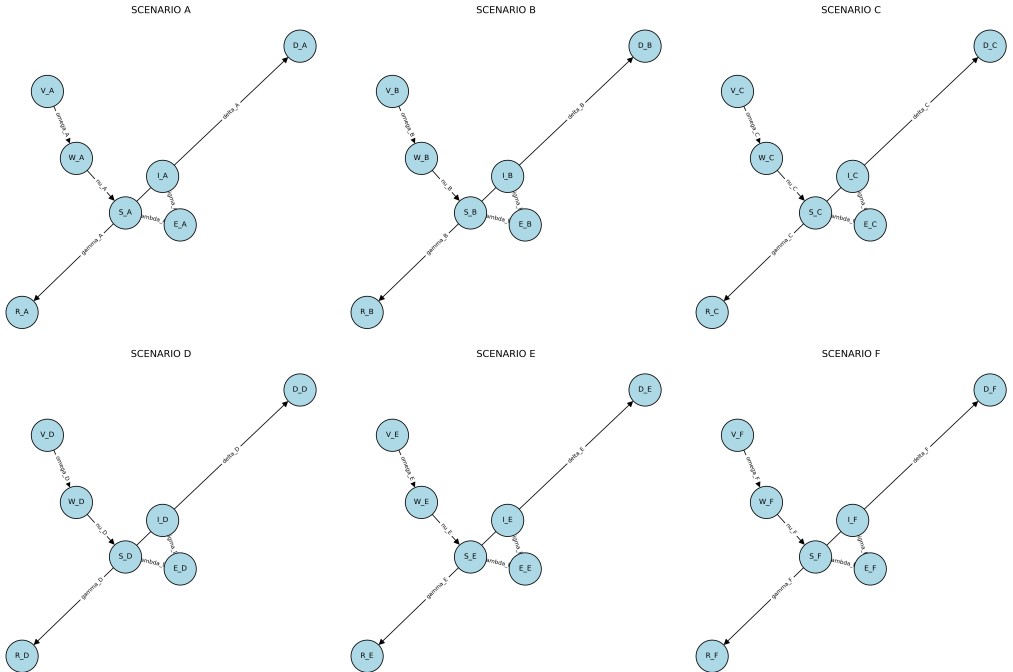

*Figure 10.* Scenario-based epidemiological flow graph construction **without external knowledge**, which leads to *oversimplification* and generates almost the *same graphs for all scenarios*

*Table 2.* Summary of notations

| Symbol | Description |
|---|---|
| $s_i$ | Scenario specification |
| $\mathcal{D}$ | Observed epidemiological data (infections, deaths) |
| $p$ | patch (spatial or age groups) |
| $t$ | Discrete time index (weeks) |
| $G = (V, E)$ | Epidemiological flow graph with compartments $V$ and transitions $E$ |
| $x \in V$ | Disease compartment (e.g., $S, E, I, R, V, W$) |
| $(u, v) \in E$ | Directed transition from compartment $u$ to $v$ with weight of disease parameter |
| $\mathcal{P}_{graph}, \mathcal{P}_{code}$, | Prompt to generate flow graph & code |
| $\mathcal{C}_g$ | Generated executable simulator corresponding to $G$ |
| $\theta^*$ | Calibrated parameter estimates after training |
| $f_{\theta^*}^s(\cdot)$ | Epidemiological simulator |
| $\mathcal{L}(\cdot)$ | MSE loss over observed trajectories |
| $\mathcal{A}_{plan}$ | LLM-based planner agent for code generation |
| $\mathcal{K}$ | Retrieved epidemiological knowledge corpus (RAG) |
| $RAG(\cdot)$ | FAISS-based retrieval operator |
| $\mathcal{C}$ | Set of epidemiological and scenario constraints |
| $\mathcal{CK}$ | Skeleton Code for LLM Planner Agent |

```
# New infections
new_inf = beta * S * I * self.dt
new_inf = torch.min(new_inf, S)  # cannot infect more than susceptible

# Recoveries
recov = gamma * I * self.dt
recov = torch.min(recov, I)

# Deaths
deaths = ifr * recov  # fraction of recovered that die
deaths = torch.min(deaths, I - recov)  # deaths cannot exceed remaining infected after recoveries
# To keep simple, assume deaths occur at recovery time
deaths = ifr * recov

# Update compartments
S = S - new_inf + waning * R * self.dt  # waning immunity returns R to S
I = I + new_inf - recov - deaths
R = R + recov - waning * R * self.dt
D = D + deaths

# Update cumulative infections
C = C + new_inf
```

*Figure 11.* Simulator well fitted, but mechanistic equation invalid. Death is not a fraction of recovered

low error alone is not reliable.

2. **Main novelty is the epidemiological flow graph.** A critical challenge in epidemic modeling is that a system can have a high numerical fit to observed trajectories while encoding a fundamentally incorrect causal structure—such as omitting waning immunity or suppressing reinfection. Such models are unsuitable for counterfactual reasoning or real-world policy guidance, yet these structural failures are often latent and cannot be identified through trajectory fitting alone (Fig 4b).

3. By verifying the flow graph against domain knowledge before code generation, EpiAgent ensures compartmental correctness by construction. Skipping this step yields invalid projections in $> 87\%$ of runs (§6.1). We also manually checked a subset of generated flow graphs with epidemiologists, who confirmed that they captured vaccination eligibility, waning, and immune escape correctly.

**Flow-Graph Versus Hard-Coded Constraints, Skeleton Code, and Verifier Feedback** Here, correctness measures the fraction of generated models that are valid across scenario differentiation, compartment flow conservation, and PyTorch differentiability. Removing the flow graph caused most models to retain incorrect transitions despite verifier feedback; removing verifier feedback exposed scenario-parameter errors (especially in immune escape handling); and removing hardcoded constraints reduced overall correctness mainly due to flow-conservation violations.

**Scaffolded Autonomy in EpiAgent** EpiAgent uses fixed components like structural constraints and a hardcoded skeleton, but this is a deliberate design choice mirroring how human experts work. Rather than redefining foundational principles for each new disease, epidemiologists build on established field knowledge and tailor model structure to the conditions of the specific scenario being studied.

By hardcoding theoretical foundations derived from established literature, the framework remains autonomous where it matters most: automating the selection of compartments and transitions for a given scenario. The skeleton code and optimization setup are minimal and only fix the execution interface, not the epidemic model structure. The actual bottleneck—designing the correct compartmental structure for new variants or policy changes—is what EpiAgent automates end-to-end.

The framework can generalize model formulation for epidemic scenarios beyond SARS-CoV-2 to any setting with a natural-language description and observational data. For example, applying it to flu forecasting requires only changing the disease name and target variables in the prompt. Any additional domain knowledge can be retrieved directly from epidemiological literature via RAG, ensuring the model strictly follows all structural constraints regardless of the disease or scenario.

### C.1. Robustness Under Incomplete Scenario Descriptions

When scenario descriptions are ambiguous, the system tends to generate conservative, simpler, more canonical structures rather than hallucinating plausible-but-wrong mechanisms.

**Counterfactual Relationship Inference** Without explicit scenario-relationship guidance, EpiAgent creates independent simple simulators $f_{\theta_s}^s$ per scenario rather than a single shared-parameter model $f_{\theta}^s$. This confirms that while the flow graph ensures structural correctness within each scenario, inferring counterfactual relationships across scenarios $\mathcal{S}$ requires explicit specification in the prompt.

**Partial Encoding of Scenarios** The failure mode under ambiguous descriptions is under-specification (e.g., missing waning or vaccination strata) rather than structurally

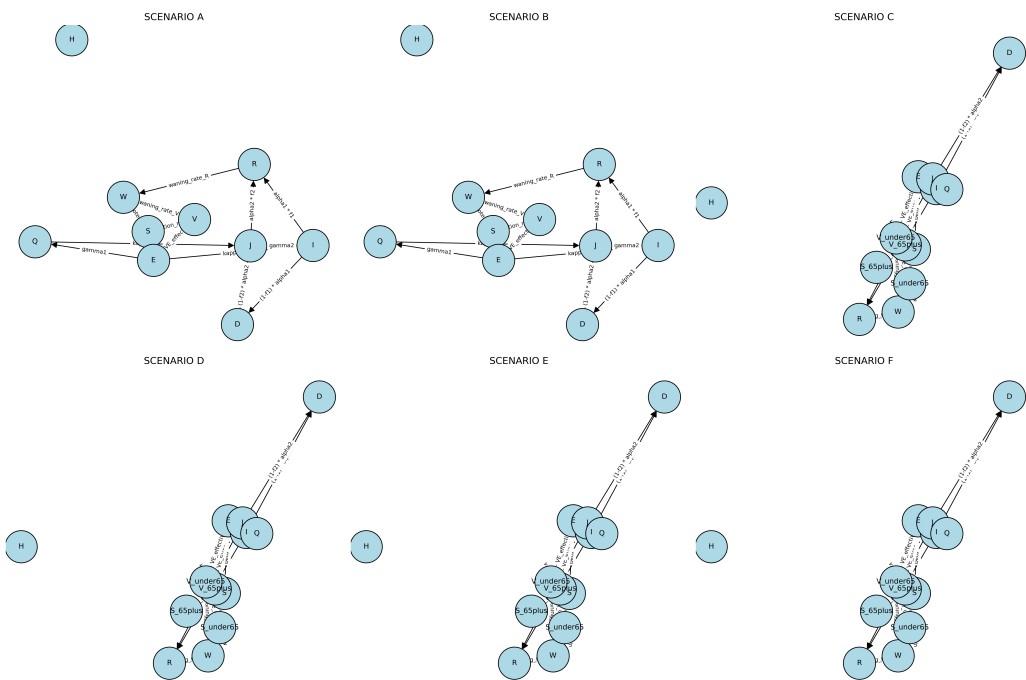

*Figure 12.* If **no constraints** are written, the graph generation model will generate the *most complicated flow dynamics*, introducing unnecessary states which are not relevant to the scenarios

```python
# -----------------------------------------------------------------
# Minimal skeleton for a differentiable epidemic simulator.
# You must fill in ALL model logic.
# -----------------------------------------------------------------
class mecha_model(nn.Module):
    def __init__(self, T, patches=56, scenario_id="A"):
        super(mecha_model, self).__init__()
        ## HERE, T IS IN WEEKS
        self.T = T
        self.patches = patches #USA_states as patches
        self.scenario = scenario_id

        # Define the trainable disease parameters here, which are shared
across all scenarios.
        # -----------------------------------------------------------------
        |
    def get_scenario_parameters(self, scenario_id):
        # Return any scenario-specific variable needed
        raise NotImplementedError

    def forward(self) -> Tuple[torch.Tensor]:
        device = "cuda"

        # Initialize all disease_state variables as fractions in [0, 1]

        M = []    # cumulative deaths per week (patches × T)
        Inf = [] # cumulative infections per week (patches × T)

        # Retrieve scenario parameters
        # params = self.get_scenario_parameters(self.scenario)

        for t in range(self.T):
            # Implement disease_state updates using explicit dt
            # Ensure numerical stability and differentiability

            M.append(D.clone())
            Inf.append(C.clone())

        return torch.stack(M).T, torch.stack(Inf).T #(patches × T)
```

*Figure 13.* Skeleton Code for Case Study I

invalid transitions—the latter are caught and rejected at the graph level. For instance, in Case Study I, without explicit scenario descriptions in prompt, for $\mathcal{S} = \{A, \ldots, F\}$, immune escape was only partially encoded: scenarios $A$ and $B$ shared identical compartmental structure with no $R \rightarrow S$ reinfection pathway, leaving high immune escape structurally absent.

## D. Comparison with Models

### D.1. EPIAGENT vs General-Purpose LLM Agents

EpiAgent was compared against general-purpose LLM agent frameworks as baselines, including MLGym and HDTwinGen, but they were found unsuitable for epidemiological modeling:

- **MLGym** (Nathani et al.): Defaults to naive SEIRD models. Despite explicit prompting, it lacks the domain architecture to handle complex features like vaccination policies or immune escape.

- **HDTwinGen** (Holt et al.): Produces structurally incorrect models with non-meaningful state flows that violate core mechanistic logic.

Ultimately, these systems prioritize functional code over scientific validity and often fail to enforce the biological and physical constraints.

| **Verification Steps** | **Scenario Output Validation** |
|---|---|
| #### **Key Assumptions**
- Compartments and flows are as specified in the scenario.
- All rates are non-negative and shared across scenarios except for scenario-specific multipliers.
- Initial conditions are the same for all patches and scenarios.
- No seasonality, no demographic change, no stochasticity.
- Age split is fixed at 18% for 65+.

#### **Verification Steps**
- Check that all flows match the scenario specification.
- Check that all rates are non-negative.
- Check that cumulative deaths and infections are non-decreasing.
- Check that the sum of all compartments does not exceed 1 (or N).
- Check that scenario-specific flows (e.g., vaccination only in 65+ in C/D) are implemented as specified.
- Check that outputs (peak, growth rates, burden) are plausible and consistent with scenario differences.

—
### 4. **Explicit Constraints from Input Scenarios**

- **No negative rates** (already enforced by squaring).
- **No vaccination in S->V for scenarios A/B**.
- **Vaccination only in 65+ for C/D, in all for E/F**.
- **Immune escape multipliers only affect beta and beta_w**.
- **No new variants except as modeled by immune escape multipliers**.
- **No NPIs or reactive interventions**.
- **Waning rates must be within plausible range (median 3-10 months)**.
- **Initial conditions must be consistent across scenarios**. | #### **a. Output Validation for Each Scenario**

- **Scenario A (no booster, low immune escape):**
 - Peak infections: 18.2, final deaths: 0.21
 - **Validation**: Reasonable, as this is the baseline scenario.

- **Scenario B (no booster, high immune escape):**
 - Peak infections: 39.4, final deaths: 0.49
 - **Validation**: Much higher burden, as expected with high immune escape. However, the loss is higher. This may be due to the immune escape multiplier being too simplistic or not capturing the true effect.

- **Scenario C (65+ booster, low immune escape):**
 - Peak infections: 9.2, final deaths: 0.13
 - **Validation**: Lower burden than A, as expected due to 65+ booster. |

(second subcolumn of right box):

- **Scenario D (65+ booster, high immune escape):**
  - Peak infections: 24.5, final deaths: 0.31
  - **Validation**: Higher than C, lower than B, as expected.

- **Scenario E (all booster, low immune escape):**
  - Peak infections: 3.7, final deaths: 0.065
  - **Validation**: Lowest burden, as expected.

- **Scenario F (all booster, high immune escape):**
  - Peak infections: 9.2, final deaths: 0.14
  - **Validation**: Higher than E, lower than D/B, as expected.

#### **b. What to Fix if Validation Fails**
- **Scenario B**: If the model may not be capturing the effect of high immune escape well. Consider:
  - Allowing immune escape multipliers to affect more than just beta/beta_w (e.g., waning rates).
  - Adding a scenario-specific parameter for immune escape that can be tuned.
  - Adding seasonality to better fit late growth/decline.

#### **c. How Outputs Should Differ Across Scenarios**

| Scenario | Should differ by | What to check in output | What is in the output | Is consistent? |
|----------|------------------|-------------------------|-----------------------|---------------|
| A | Baseline | Moderate peak, moderate deaths | Peak: 18.2, Deaths: 0.21 | Yes |
| B | Higher immune escape | Higher peak, higher deaths | Peak: 39.4, Deaths: 0.49 | Yes |
| C | 65+ booster | Lower deaths than A, lower peak | Peak: 9.2, Deaths: 0.13 | Yes |
| D | 65+ booster, high escape | Higher than C, lower than B | Peak: 24.5, Deaths: 0.31 | Yes |
| E | All booster | Lowest peak and deaths | Peak: 3.7, Deaths: 0.065 | Yes |
| F | All booster, high escape | Higher than E, lower than D/B | Peak: 9.2, Deaths: 0.14 | Yes |

*Figure 14.* Verification and scenario-output validation in EPIAGENT. Left: Explicit verification steps and hard constraints enforced on synthesized epidemiological simulators, including mechanistic validity, parameter constraints, and scenario compliance. Right: Scenario-wise output validation illustrating expected qualitative ordering of peak infections and cumulative deaths across counterfactual scenarios, along with diagnostic guidance when validation fails.

## **Actionable Feedback for Next Iteration**

1. **Add seasonality** to the infection rate to better fit late growth/decline and improve fit across all scenarios.
2. **Allow initial conditions to be patch-specific** (or at least scenario-specific) and possibly trainable, to better match observed data.
3. **Check and enforce population conservation** (sum of all compartments ≤ 1) at each step.
4. **Allow the 65+ fraction to be patch-specific** if data supports it.
5. **Consider more flexible immune escape modeling** (e.g., allow immune escape to affect waning rates or other parameters, or make the multiplier weakly trainable).
6. **Review the implementation of all-ages booster in E/F** to ensure vax_rate_u65 is sufficient and correctly applied.
7. **If some parameters are consistently near zero, consider fixing or removing them** to improve identifiability and interpretability.

*Figure 15.* Actionable feedback generated by the agentic refinement loop in model generation phase.

*Table 3.* Impact of language model capacity, structural priors, and constraints on the quality of automatically generated epidemiological simulators.

| Factor | High-Capacity Models (GPT-4.x) | Lightweight Models (Mini) | Key Implications |
|---|---|---|---|
| Language model capacity | Strong inductive reasoning; infers structure with limited guidance | Limited implicit reasoning; sensitive to prompt formulation | Higher-capacity models reduce but do not eliminate the need for explicit structure |
| Skeleton code availability | Helpful but not strictly required; improves stability and consistency | Critical for correctness; prevents structural and numerical errors | Skeleton code acts as a structural prior, especially important for smaller models |
| Amount of guidance required | Low to moderate (high-level objectives often sufficient) | High (explicit constraints, parameter bounds, and templates needed) | Guidance compensates for reduced model capacity |
| Behavior without constraints | Produces plausible but epidemiologically invalid or unstable dynamics | Frequently generates incoherent or numerically unstable simulators | Constraints are essential for epidemiological realism across all models |
| Overall simulator quality | High when combined with constraints and minimal scaffolding | Competitive only with strong scaffolding and constraint enforcement | Reliable model generation emerges from the interaction of all factors |

*Table 4.* Unified hard constraints enforced during flow-graph verification and simulator synthesis. Here $*$ denotes any compartment.

| Constraint Type | Specification |
|---|---|
| **Flow Validity** | $S \rightarrow R$ (no direct recovery) 
 $S \rightarrow I$ (latent period required) 
 $\{S, E, R, V, W\} \rightarrow D$ (death only from severe states) 
 $D \rightarrow *$ forbidden (terminal state) 
 $\{E, I, J, H\} \rightarrow V$ (vaccination eligibility) 
 $\{S, E, I, J, H\} \rightarrow W$ (waning semantics) 
 $\{E, I, R, J, H\} \rightarrow E$ (exposure semantics) |
| **Execution Interface** | Must complete a fixed PyTorch skeleton |
| **Differentiability** | End-to-end differentiable; no parameter reassignment in `forward()` |
| **Numerical Stability** | Explicit time steps; no NaNs or Infs |
| **State Semantics** | All states represent population fractions in $[0, 1]$ |
| **Output** | Return cumulative infections and deaths $(P \times T)$ |
| **Parameter Sharing** | Differentiable disease parameters are shared across all scenarios |
| **Scenario Isolation** | Scenario-specific variables are non-trainable |
| **Constraint Handling** | No forced non-negativity (e.g., ReLU/Softplus) |

## D.2. Evaluation Across Different LLMs

**Open-Source Models** For open-source models, three Qwen2.5-Coder variants were evaluated:

- **7B-Instruct** failed to generate executable code, consistently violating the skeleton interface and producing code that could not run in the predefined environment.

- **14B-Instruct** produced executable but overly simplistic simulators that did not reflect scenario-level differences.

- **32B-Instruct** showed performance comparable to GPT-4.1, successfully generating scenario-consistent simulators within the EpiAgent pipeline.

**Frontier Models** GPT-5.2 paired with the flow graph and V&V layer generated correct models. However, GPT-5.2 could not participate in the feedback loop—its content policy restricts biological experimentation-related responses, preventing actionable V&V feedback from being returned.

Under zero-shot prompting (no flow graph, no V&V, no feedback), both GPT-5.2 and Claude Opus 4.6 produced reasonable models, but not the epidemiologically precise simulators that EpiAgent generates. To ensure a fair comparison, the optimization function was kept identical across models.

1. **GPT-5.2.** Generated a plausible but overly complex model. Attempted RK4 but implemented incorrectly—worse than Euler. Counted $dS$ twice for reinfection—mass conservation broken at ODE level (invalid transition).

| VERIFICATION |
|---|

**REASONING**:
- **S (Susceptible):** Individuals who are susceptible to infection. This is the starting point for infection and vaccination.
- **E (Exposed):** Individuals who have been infected but are not yet infectious. This compartment captures the incubation period.
- **I (Infectious):** Individuals who can transmit the disease.
- **R (Recovered):** Individuals who have recovered and have some immunity, which wanes over time.
- **V (Vaccinated):** Individuals who have vaccine-induced immunity. In scenarios C-F, vaccination is stratified by age groups (V_65plus and V_other) to capture differential booster uptake.
- **D (Dead):** Individuals who have died from the disease.
  These compartments reflect the natural history of infection, immunity acquisition (natural and vaccine-induced), waning immunity, and mortality.

**TRANSITIONS:**
- **S -> E [beta]:** Infection of susceptibles at a rate proportional to transmission parameter beta. This is standard SEIR infection flow.
- **E -> I [sigma]:** Progression from exposed to infectious at rate sigma, representing incubation period.
- **I -> R [gamma]:** Recovery from infection at rate gamma.
- **I -> D [mu]:** Disease-induced death at rate mu.
- **R -> S [waning]:** Waning of natural immunity, returning recovered individuals to susceptible.
- **V -> S [waning_vax]:** Waning of vaccine-induced immunity, returning vaccinated individuals to susceptible.
- **S -> V [vaccination]:** Vaccination of susceptibles, moving them into vaccinated compartment.

In scenarios C-F, vaccination and vaccinated compartments are split by age group:
- **S -> V_65plus [vaccination_65plus]:** Vaccination of 65+ age group.
- **S -> V_other [vaccination_other]:** Vaccination of under 65 age group (negligible or significant depending on scenario).
- **V_65plus -> S [waning_vax]:** Waning vaccine immunity in 65+.
- **V_other -> S [waning_vax]:** Waning vaccine immunity in other age groups.
The infection rate beta is adjusted for immune escape scenarios (beta_lowIE vs beta_highIE), reflecting the increased transmissibility or immune evasion.

**ALIGNMENT**:
- The compartmental structures and transitions align well with the scenario descriptions.
- Scenarios A and B do not include vaccination transitions because no future booster is recommended; vaccination waning is modeled but no new vaccination occurs.
- Scenarios C and D include vaccination for 65+ only, with negligible vaccination for others.
- Scenarios E and F include vaccination for both 65+ and other groups.
- Immune escape is modeled by adjusting beta (infection rate) to beta_lowIE or beta_highIE.
- Waning immunity is modeled for both recovered and vaccinated compartments.
- Deaths are explicitly modeled.
- The model includes vaccination and waning for vaccine-induced immunity, consistent with the scenario specification.
- The split of vaccinated compartments by age group in scenarios C-F matches the vaccination coverage assumptions.

**ISSUES**:
- **No explicit modeling of waning immunity from natural infection leading to vaccinated compartments or vice versa:** The model assumes waning immunity returns individuals to susceptible, which is acceptable but does not capture hybrid immunity explicitly. However, this is consistent with the scenario description which leaves waning and immune escape implementation to teams' discretion.
- **No explicit modeling of breakthrough infections in vaccinated compartments:** The model does not show transitions from V or V_65plus/V_other to E or I. This could be implicit in the infection rate beta, but typically vaccinated individuals have reduced susceptibility. This is a simplification but common in compartmental models.
- **No explicit modeling of booster timing or vaccine efficacy decay over time:** The model uses vaccination compartments and waning but does not explicitly model reformulated booster availability dates or VE decay over time. This is likely handled in parameter values rather than structure.
- **No explicit age stratification beyond vaccination compartments:** The model only stratifies vaccinated compartments by age group but not other compartments. This is a simplification but acceptable given the scenario focus on vaccination coverage by age.
- **No explicit hospitalization compartment:** The scenarios require projections of hospitalizations, but the model does not include a hospitalization compartment. This may be handled outside the compartmental structure (e.g., via post-processing or separate modeling).
- **No explicit variant compartments:** Immune escape is modeled via beta adjustment, but no explicit variant compartments are included, consistent with scenario instructions.
*VERDICT: VALID*

*Figure 16.* Graph Verification checks and reasoning while generating the flow to ensure consistency between the scenarios

*Table 5.* Ablation of EpiAgent Components.

| Configuration | Correctness (%) | Main Issues |
|---|---|---|
| w/o hard-coded constraints | ∼69% | Missing depletion flows, PyTorch errors |
| w/o verifier feedback | ∼53% | Structurally correct but no error recovery |
| w/o flow graph | ∼42% | Invalid transitions, missing waning flows |

2. **Claude Opus 4.6.** Valid transitions but introduced extra states not specified in the scenario (e.g., hospitalization). Euler only but correctly implemented; did not double count as GPT-5.2. The initial condition was specified to be static, but this violates the skeleton spec—a design error not an implementation error. However, the overall projections do not reflect the intended scenario differences observed in the ensemble forecasts.

These results suggest that, even with the latest models, the full EpiAgent pipeline remains necessary for reliable deployment.

## E. Soundness of the Framework

**Feedback Loops** Feedback iterations progressively resolved failures from structural to implementation level: 17% correct generations at iteration 1 (scenario generation empty/crashing), 50% at iteration 2 (incorrect vaccination), 67%–83% at iterations 3–4 (flow conservation violations), and 100% by iteration 5.

**Human Error Checks** Common failure modes can be checked at two levels:

1. **Model-Level Checks.** Verify that the compartmental structure is correct, e.g., valid SEIRD flows; all required compartments are present, transitions are epidemiologically meaningful, population is conserved; whether the model captures the scenario-specific assumptions appropriately. For example, even if vaccination is not represented through an explicit $V$ compartment, it should still be clear that its effect is handled correctly in the model dynamics.

2. **Projection-Level Checks.** Verify that the outputs are consistent with epidemiological expectations and scenario definitions. For instance, projections should reflect that vaccination reduces infections and deaths, and that broader booster coverage would generally have a larger effect than booster coverage limited to high-risk groups, which in turn should outperform a no-booster setting.

```
def get_scenario_parameters(self, t):
    # Accept both scenario_id and short scenario names
    sid = self.scenario
    # Accept both full and short scenario names
    if sid in ["A-2023-04-16", "noBoo_lowIE", "A"]:
        beta = self.log_beta[t].exp()
        sigma = self.log_sigma[t].exp()
        gamma = self.log_gamma[t].exp()
        return {"beta": beta, "sigma": sigma, "gamma": gamma}
    elif sid in ["B-2023-04-16", "noBoo_highIE", "B"]:
        beta = self.log_beta[t].exp()
        sigma = self.log_sigma[t].exp()
        gamma = self.log_gamma[t].exp()
        return {"beta": beta, "sigma": sigma, "gamma": gamma}
    elif sid in ["C-2023-04-16", "65Boo_lowIE", "C"]:
        beta = self.log_beta[t].exp()
        sigma = self.log_sigma[t].exp()
        gamma = self.log_gamma[t].exp()
        booster_65p = self.log_booster_65p[t].exp()
        omega = self.log_omega[t].exp()
        beta_v = self.log_beta_v[t].exp()
        return {"beta": beta, "sigma": sigma, "gamma": gamma, "booster_65p": booster_65p, "omega": omega, "beta_v": beta_v}
    elif sid in ["D-2023-04-16", "65Boo_highIE", "D"]:
        beta = self.log_beta[t].exp()
        sigma = self.log_sigma[t].exp()
        gamma = self.log_gamma[t].exp()
        booster_65p = self.log_booster_65p[t].exp()
        omega = self.log_omega[t].exp()
        beta_v = self.log_beta_v_highIE[t].exp()
        return {"beta": beta, "sigma": sigma, "gamma": gamma, "booster_65p": booster_65p, "omega": omega, "beta_v": beta_v}
    elif sid in ["E-2023-04-16", "allBoo_lowIE", "E"]:
        beta = self.log_beta[t].exp()
        sigma = self.log_sigma[t].exp()
        gamma = self.log_gamma[t].exp()
        booster_all = self.log_booster_all[t].exp()
        omega = self.log_omega[t].exp()
        beta_v = self.log_beta_v[t].exp()
        return {"beta": beta, "sigma": sigma, "gamma": gamma, "booster_all": booster_all, "omega": omega, "beta_v": beta_v}
    elif sid in ["F-2023-04-16", "allBoo_highIE", "F"]:
        beta = self.log_beta[t].exp()
        sigma = self.log_sigma[t].exp()
        gamma = self.log_gamma[t].exp()
        booster_all = self.log_booster_all[t].exp()
        omega = self.log_omega[t].exp()
```

*Figure 17.* Improper prompting leads to partial encoding of scenarios

## F. Improvement and Future Work

The primary next step is full automation: given a published epidemiological paper, EpiAgent would automatically extract the scenario, construct the model, and produce comparable results without manual prompt engineering. Case Study II partially demonstrates this—EpiAgent instantiated an established behavioral SEIR model from literature—but scenario description and prompts were still manually constructed. Automating this extraction is the immediate priority.

A second direction is fine-tuning the LLM for epidemiological flow graph generation using curated examples from epidemiological literature, which would improve graph quality and reduce verification iterations for complex scenarios.

Additional directions include convergence-aware stopping criteria to address late-stage regression, spatially-structured metapopulation extensions, and Bayesian uncertainty quantification for policy use.

**Applicability Beyond COVID-19** EpiAgent can be used on FluForecasting data with only minor prompt modifications (as the task changes from projection to forecasting). Unlike SMH, this is a forecasting task, so there are no multiple coupled scenario cases.[1]

**Model Complexity** The models formulated here incorporate age-structured vaccination and booster effects, differential immune escape, waning immunity, disease-induced deaths, and imperfect vaccination. These are not simple extensions of baseline epidemic models, rather they involve

---

[1] https://github.com/cdcepi/
FluSight-forecast-hub/tree/main

```
class DifferentiableSIRDModel(nn.Module):
    """
    SIRD compartmental ODE implemented in a differentiable, functional way.
    No in-place operations. All states are returned as new tensors per time step.
    """
    def __init__(self, n_regions, n_timesteps, population):
        super().__init__()
        self.n_regions = n_regions
        self.n_timesteps = n_timesteps
        self.log_beta = nn.Parameter(torch.zeros(n_regions))   # infection rate
        self.log_gamma = nn.Parameter(torch.zeros(n_regions))  # recovery rate
        self.log_mu = nn.Parameter(torch.zeros(n_regions))     # mortality rate
        self.register_buffer("population", population)

    def forward(self, I0, R0, D0):
        n, T = self.n_regions, self.n_timesteps
        beta = torch.nn.functional.softplus(self.log_beta)
        gamma = torch.nn.functional.softplus(self.log_gamma)
        mu = torch.nn.functional.softplus(self.log_mu)
        S = []
        I = []
        R = []
        D = []
        S0 = self.population - I0 - R0 - D0
        s, i, r, d = S0, I0, R0, D0
        S = []
        I = []
        R = []
        D = []
        S0 = self.population - I0 - R0 - D0
        s, i, r, d = S0, I0, R0, D0
        for t in range(T):

            S.append(s)
            I.append(i)
            R.append(r)
            D.append(d)
            new_infect = beta * s * i / self.population
            new_recover = gamma * i
            new_death = mu * i
            s = s - new_infect
            i = i + new_infect - new_recover - new_death
            r = r + new_recover
            d = d + new_death
        S = torch.stack(S, dim=1) # [n, T]
        I = torch.stack(I, dim=1)
        R = torch.stack(R, dim=1)
        D = torch.stack(D, dim=1)
        cumulative_M = D
        cumulative_I = I0.unsqueeze(1) + (self.population.unsqueeze(1) - S)
        return cumulative_M, cumulative_I
        return cumulative_M, cumulative_I
```

*(a)* MLGym: cannot reflect different scenarios

```
# New infections
new_inf = beta * S * I * self.dt
new_inf = torch.min(new_inf, S)  # cannot infect more than susceptible

# Recoveries
recov = gamma * I * self.dt
recov = torch.min(recov, I)

# Deaths
deaths = ifr * recov  # fraction of recovered that die
deaths = torch.min(deaths, I - recov)  # deaths cannot exceed remaining infected after recoveries
# To keep simple, assume deaths occur at recovery time
deaths = ifr * recov

# Update compartments
S = S - new_inf + waning * R * self.dt  # waning immunity returns R to S
I = I + new_inf - recov - deaths
R = R + recov - waning * R * self.dt
D = D + deaths

# Update cumulative infections
C = C + new_inf
```

*(b)* HDTwinGen: made impossible transitions

*Figure 18.* Examples of naive generated models from MLGym and HDTwinGen containing incorrect flows.

multiple interacting mechanisms that jointly shape epidemic dynamics. Accounting for these processes in a coherent framework substantially increases both the structural and analytical complexity of the model.

## G. Role of Agents in EPIAGENT

EPIAGENT relies on a collection of specialized agents, each responsible for a distinct stage of the epidemic simulator construction and validation pipeline. Together, these agents ensure that the generated models are structurally correct, epidemiologically meaningful, and executable.

In particular, agents are used for the following tasks:

1. **Flow-graph construction.** Given a natural-language scenario description and retrieved epidemiological knowledge, an agent constructs a compartmental flow graph representing disease states and admissible transitions. This graph encodes the high-level mechanistic structure implied by the scenario, such as vaccination

policies, waning immunity, immune escape, and behavioral effects.

2. **Flow-graph verification.** The constructed flow graph is evaluated by a dedicated verification agent. This agent enforces hard epidemiological constraints (e.g., valid causal transitions, conservation of population mass, and absence of impossible flows such as direct $S \to R$ recovery). Graphs that violate these constraints are rejected, and structured error feedback is returned to guide regeneration.

3. **Model instantiation and calibration.** Once a flow graph is verified, it is passed to an LLM-based planner agent, which translates the graph into a system of ordinary differential equations and corresponding executable simulator code. The resulting simulator is then calibrated against real-world epidemiological data using gradient-based optimization or sampling-based inference.

4. **Code-level verification and validation.** The generated simulator code is further analyzed by a set of agents responsible for execution-time verification and validation. These agents check for numerical instability, invalid state evolution (e.g., negative compartment values), violations of population conservation, and non-monotonic cumulative quantities. Feedback from these agents is used to iteratively correct the code when necessary.

Through this multi-agent design, EPIAGENT enforces correctness at multiple levels: structural (flow graph), dynamical (ODE system), and executable (simulation behavior). The distinct roles of these agents, including the LLM-based planner, are summarized in Table 6.

In Figure 14 and 15, we illustrate representative examples of the checks performed during the verification process. These examples highlight the types of structural and execution-level constraints enforced by the agents; however, the verification is not limited to these cases, as the exact checks invoked may vary due to stochastic generation and scenario-specific model structure.

## H. Discussion

**Impact of Language Model Choice.**   Our results indicate that the quality and reliability of the generated epidemiological simulators are strongly influenced by the choice of underlying language model. Larger and more capable models, e.g., GPT-4.1, in comparison to GPT-4.1 mini, GPT-4-0, GPT-4-0-mini, consistently produce simulator code that better adheres to epidemiological structure and scenario semantics, even under relatively weak prompting. In contrast, smaller or lightweight models rely on the skeleton code

and are more prone to numerical instability or violations of epidemiological constraints when operating without explicit guidance. We summarize this observation in Table 3 and 8 .

**Role of Skeleton Code and Structural Priors.**   Across all model variants, the presence of a skeleton codebase improves generation quality. Skeleton code acts as a strong structural prior that constrains the search space of possible programs, reducing ambiguity in both model dynamics and data flow. While larger models are sometimes able to infer missing structure implicitly, smaller models rely heavily on such explicit framing to avoid degenerate or incoherent simulator implementations. This suggests that skeleton code and language model capacity play complementary roles in simulator generation. We show a sample skeleton code for Case Study I in Figure 13.

**Effect of Constraints on Generated Simulators.**   When constraints are removed or weakened, all model variants exhibit degraded performance, though the failure modes differ. Larger models tend to violate epidemiological realism (e.g., implausible parameter values or unstable long-term dynamics), while smaller models more frequently produce structurally invalid simulators or numerically unstable updates. Explicit constraint enforcement is therefore essential not only for epidemiological correctness but also for maintaining robustness across different language model capacities.

Table 4 summarizes the hard structural and execution constraints enforced throughout model synthesis. These constraints restrict allowable compartmental transitions to epidemiologically valid flows (e.g., enforcing latent infection, terminal death states, and semantically correct vaccination and waning pathways) while simultaneously constraining simulator execution to a fixed, differentiable PyTorch interface with stable numerical updates. By fixing parameter sharing, scenario isolation, and optimization behavior, the framework ensures that candidate simulators differ only in their structural mechanisms rather than optimizer tuning or re-fitting artifacts.

**Structural Revision over Parameter Clipping.**   Imposing hard constraints, such as $ReLU$ activations on disease parameters, can mask underlying model misspecification. In early iterations, the optimizer frequently drives parameters to negative values when the compartmental logic is incompatible with the ground truth data. By treating these boundary violations as signals for topological revision rather than simple optimization constraints, the VnV agent forces a re-evaluation of the model structure. This ensures that non-negativity is a property of a correct model rather than a forced numerical artifact.

*Table 6.* Agent responsibilities and guarantees in the VnV-driven generation framework. Each agent enforces a distinct modeling aspect—ranging from mathematical correctness to scenario plausibility—ensuring that successive generations improve structural validity, interpretability, and scenario fidelity rather than only minimizing validation loss.

| Agent | Primary Function | Aspect | Outputs Produced | How This Improves Model Generation |
|---|---|---|---|---|
| **Verification Agent** | Evaluates hard mathematical and epidemiological constraints on simulator states and parameters | **Correctness** (legal dynamics, numerical validity) | Binary verdict (`PASS`/`FAIL`); violation-specific error logs (e.g., negative rates, mass imbalance) | Prevents invalid or unphysical models from entering optimization or interpretation stages; constrains the search space to epidemiologically admissible simulators |
| **Validation Agent** | Assesses whether verified outputs are scientifically meaningful and consistent with scenario intent | **Plausibility & Scenario Fidelity** (realistic dynamics, intervention effects) | Scenario-level validation status (`PASS`/`WARN`/`FAIL`); diagnostics such as scenario collapse or missing mechanisms | Detects structurally insufficient models that achieve low loss but fail to express intended scenario differences; forces mechanistic adequacy beyond curve fitting |
| **Reasoning Agent** | Interprets logs and optimized parameters to identify structural causes of failure | **Interpretability & Diagnosis** | Mechanistic explanations linking observed failures to missing compartments, flows, or constraints | Prevents repeated generation of equivalent white-box models; explains why loss minimization succeeded or failed in epidemiological terms |
| **Feedback Agent** | Converts reasoning diagnostics into concrete, actionable regeneration directives | **Learnability Across Generations** | Explicit structural modification instructions (e.g., add vaccination compartments, separate waning pathways) | Guides the LLM planner toward targeted architectural changes, accelerating convergence toward valid and expressive models |
| **LLM Planner Agent** | Synthesizes feedback with prior generation history to design the next simulator specification | **Structured Exploration** | Revised model blueprint specifying compartments, parameters, and scenario logic | Ensures iterative improvement occurs at the model-structure level, not merely via parameter re-optimization |

## I. Design Choices for Disease Parameterization

A key design decision in EPIAGENT concerns how disease dynamics are parameterized within the generated simulators. We consider three increasingly expressive modeling choices—*time-invariant*, *time-varying*, and *neural-augmented* dynamics—each trading off interpretability, flexibility, and data efficiency.

**Time-invariant mechanistic models.** In the simplest setting, disease parameters are assumed constant over time. Let $x(t) \in \mathbb{R}_+^K$ denote the compartmental state vector induced by the verified flow graph. The epidemic dynamics are governed by a system of ordinary differential equations

$$\frac{dx(t)}{dt} = f_{\text{mecha}}(x(t); \theta),$$

where $\theta = \{\beta, \sigma, \gamma, \ldots\}$ are fixed disease parameters such as transmission, incubation, and recovery rates. This formulation corresponds to classical compartmental models and favors interpretability and robustness. In EPIAGENT, this option is selected by default unless the scenario specification explicitly requires temporal adaptation.

**Time-varying parameter models.** To capture non-stationary dynamics induced by interventions, behavior change, or seasonal effects, EPIAGENT optionally allows disease parameters to vary over time. In this setting, the dynamics take the form

$$\frac{dx(t)}{dt} = f_{\text{mecha}}(x(t); \theta(t), \phi(s)),$$

where $\theta(t)$ denotes time-dependent parameters (e.g., $\beta(t)$ or $\gamma(t)$) and $\phi(s)$ encodes scenario-specific controls. Time variation may be implemented through parametric functions or externally provided covariates (e.g., mobility or seasonality). This design improves short-term fit while preserving the underlying mechanistic structure. In particular, when the observed data exhibit multiple epidemic waves or recurrent peaks, time-varying disease parameterization enables the model to adapt transmission dynamics over time and achieve substantially improved fidelity compared to time-invariant formulations.

**Neural-augmented dynamics.** When additional flexibility is required, EPIAGENT supports augmenting mechanistic models with neural components. Specifically, a neural function $g_\psi(\cdot)$ is introduced to model residual or latent effects:

$$\frac{dx(t)}{dt} = f_{\text{mecha}}(x(t); \theta) + g_\psi(x(t), t),$$

*Table 7.* Verification, validation, and feedback assessment of the final generated epidemic simulator. Verification enforces hard correctness constraints, validation ensures scientific plausibility and scenario fidelity, and feedback identifies concrete structural improvements for subsequent model generations.

| Agent | What Was Checked | Status | Implication / Actionable Outcome |
|---|---|---|---|
| **Verification** | • Non-negativity of all compartments ($S, E, I, R_{nat}, R_{vac}, D, C$) 
 • Conservation of population mass (excluding deaths) 
 • Monotonicity of cumulative infections and deaths 
 • Non-negativity of disease parameters ($\beta, \gamma, \mu, \sigma$, waning) 
 • Numerical stability across all scenarios | **PASS** | Confirms epidemiological correctness and mathematical validity; prevents invalid or unphysical simulators from being considered regardless of validation loss |
| **Validation** | • Scenario fidelity across A–F (immune escape, booster coverage) 
 • Cross-scenario ordering of infections and deaths 
 • Presence of meaningful intervention effects (boosters reduce burden) 
 • Relative severity of high vs. low immune escape scenarios | **PASS** | Demonstrates scientific plausibility and correct expression of scenario mechanisms beyond curve fitting; confirms that scenarios differ for epidemiologically meaningful reasons |
| **Feedback** | • Sensitivity of effective transmission ($R_t$) to scenario mechanisms 
 • Identification of missing couplings (vaccination, immune escape, transmission) 
 • Assessment of structural limitations despite correct outputs | **ACTIONABLE** | Triggers targeted structural improvements for the next generation, including gradual vaccination uptake, waning vaccine efficacy, age-specific mortality, seasonal forcing, and scenario-dependent transmission |
| **Decision** | Joint assessment of correctness, plausibility, and scenario expressiveness using VnV and feedback agents | **ACCEPTED** | Model is epidemiologically valid and scenario-consistent, while VnV-driven feedback provides clear guidance for extending expressiveness and improving realism in subsequent generations |

*Table 8.* Effect of language model capacity, skeleton code, and constraints on simulator generation quality. "++": strong, "+": moderate, "–": poor.

| Factor | GPT-4.x | Mini | Implication |
|---|---|---|---|
| Reasoning ability | ++ | + | Capacity matters |
| Skeleton code | + | ++ | Structure is critical |
| Guidance | + | ++ | Guidance compensates capacity |
| Stable without constraints | + | – | Constraints essential |
| Epidemiologically realistic output | + | + | Depends on scaffolding |
| Reliable simulator generation | ++ | + | Requires all factors |

where $g_\psi$ is parameterized by a neural network with parameters $\psi$. This hybrid formulation allows the simulator to learn complex, time-varying effects while retaining an interpretable mechanistic backbone. To avoid over-parameterization, neural augmentation is only introduced when explicitly requested or when simpler parameterizations fail to achieve adequate fit. As illustrated in Figure 21, neural augmentation can substantially improve fit; however, since our primary focus is on mechanistic disease modeling rather than purely predictive forecasting, we do not employ these models in the main experimental evaluation. Such neural-augmented formulations are more appropriate for forecasting-oriented tasks.

**Selection and calibration.** The choice among these parameterizations is guided by the scenario description and empirical performance. Unless specified otherwise, EPIA-GENT prioritizes time-invariant models to recover dominant epidemic trends without unnecessary complexity. More expressive parameterizations are introduced incrementally when required to improve calibration fidelity. Generated simulator code and corresponding fits are reported alongside projections to illustrate the effect of each design choice.

## J. Sensitivity to Language Model Capacity

We analyze the effect of language model capacity on simulator quality (Table 8, 3, Appendix). Larger models more reliably infer epidemiological structure with weaker guidance, while smaller models depend heavily on skeleton code and explicit constraints. However, across all model sizes, the combination of structural priors, verification, and feedback is essential for producing reliable simulators.

This suggests that robustness emerges not from language model capacity alone, but from the interaction between agentic control, structural constraints, and learning.

## K. Scenario Specification and Structural Constraints for Behavioral Epidemic Baselines

**Scenario Specification** Each experiment is defined by a scenario consisting of (i) location-specific epidemiological inputs and (ii) a behavioral mechanism choice. Location inputs include age-structured contact matrices, population by age, daily incident deaths (and optionally cases), Google mobility time series, hemisphere indicator for seasonality, and fixed age-specific infection fatality ratios (IFR). The behavioral mechanism specifies one of three PNAS baselines:

1. DDB (Data-Driven Behavior): Exogenous modulation of transmission using mobility-derived contact reductions.

2. CBF (Compartmental Behavior Feedback): Endogenous risk-averse behavior modeled via an additional susceptible compartment with transitions driven by observed deaths.

3. EFB (Effective Force-of-Infection Damping): Implicit behavioral response that nonlinearly damps transmission as a function of recent and cumulative deaths.

Scenarios do not contain equations or code; they explicitly specify assumptions and data contracts.

**Prompt Construction and Structural Constraints** From the scenario, EPIAGENT constructs a prompt that enforces a fixed execution interface and hard scientific constraints. The prompt requires the LLM to generate a single-location, age-structured SEIR-type simulator with daily time steps, causal compartment flows $S \rightarrow E \rightarrow I \rightarrow R$, and deaths derived via fixed IFR. Transmission is defined through the force of infection $\lambda(k,t) = s(t) \sum_{k'} C_{kk'} \frac{I_{k'}(t)}{N_{k'}}$, with behavior affecting $\lambda(k,t)$ according to the selected baseline. To preserve epidemiological semantics, the prompt distinguishes trainable parameters (e.g., $R_0$, incubation and removal rates, detection rate, behavioral response strengths) from fixed inputs (contact matrices, populations, IFR, mobility series, seasonality phase, reporting delay). Numerical correctness (non-negativity, mass conservation, monotonic cumulative deaths) is enforced structurally rather than via ad-hoc clamping.

## L. Limitations and Future Work

1. **Fixed Optimization Strategy.** EPIAGENT uses a Fixed Optimization Strategy, which prevents the agent from adaptively modifying learning rates during calibration. Future iterations will explore convergence-aware stopping criteria to prevent regression in model quality during later generations due to compounding noise.

2. **Iterative Code Generation Stability.** Although iterative code refinement generally improves model quality, later generations may occasionally regress due to compounding generation noise. Similarly, automatic error recovery is bounded by a fixed retry limit to ensure termination. Extending the framework with convergence-aware stopping criteria remains an important direction for future work.

---

**Algorithm 2** Iterative Verification and Validation (V&V) Agent

---

**Require:** Current Program $P_k$, Scenarios $\mathcal{S}$, Observed Data $\mathcal{D}$, Error Tolerance $\epsilon$
**Ensure:** Refined Model Structure $M_{k+1}$, Validation Status $V \in \{\text{PASS, FAIL}\}$
1: **while** $\mathcal{L}_{val} > \epsilon$ **and** $k < k_{max}$ **do**
2:     {**Phase I: Structural Verification**}
3:     Audit $P_k$ for adherence to *Brauer Compartmental Axioms*:
4:        1. **Non-negativity:** $\forall t, x_i(t) \geq 0$.
5:        2. **Mass Conservation:** $\sum_i \frac{dx_i}{dt} = 0$.
6:     {**Phase II: Scenario-Consistency Validation**}
7:     **for** each scenario $s \in \mathcal{S}$ **do**
8:        Simulate $\hat{y}_s \leftarrow \text{Exec}(P_k, s)$
9:        Evaluate $\Delta_{peak}, \Delta_{rate}$ against scenario-specific constraints $\mathcal{C}_s$.
10:        **if** $\text{Sign}(\Delta \hat{y}_s) \neq \text{Sign}(\Delta \mathcal{C}_s)$ **then**
11:          $\mathcal{F}_{log} \leftarrow$ Flag contradiction in $(V, E)$ transition logic.
12:        **end if**
13:     **end for**
14:     {**Phase III: Agentic Reasoning**}
15:     Compute Calibration Residuals: $\mathcal{R} = \|\hat{y} - \mathcal{D}\|$
16:     **if** $\text{Trend}(\mathcal{R})$ is stalled $> \tau$ **and** $P_k \in$ White-Box **then**
17:        $M_{k+1} \leftarrow P_k + \mathcal{N}_\phi(t, x)$ {Inject Neural Residual Terms, If specified in prompt}
18:        Update prompt context: $\mathcal{P} \leftarrow \mathcal{P} \oplus$ "limitations reached; initiating hybrid calibration."
19:     **else**
20:        {Maintain mechanistic interpretability but refine parameters}
21:        $M_{k+1} \leftarrow \text{RefactorLogic}(P_k, \nabla_\theta \mathcal{L}_{val}, \mathcal{F}_{log})$
22:     **end if**
23:     $k \leftarrow k + 1$
24: **end while**
25: RETURN $M_k$

---

```
            # Make disease parameters vector.
            # like there will self.T betas, self.T gammas. like ->
            self.log_beta = nn.Parameter(torch.log(torch.full((self.T,), 0.25)))
            self.sigma = nn.Parameter(torch.tensor(0.5))
            self.gamma = nn.Parameter(torch.tensor(0.1))
            self.mu = nn.Parameter(torch.tensor(0.01))
            self.nu = nn.Parameter(torch.tensor(0.005))
            self.omega_v = nn.Parameter(torch.tensor(0.001))
            self.omega_r = nn.Parameter(torch.tensor(0.0005))

            # Initial conditions
            self.S0 = nn.Parameter(torch.ones(patches) / patches)
            self.E0 = nn.Parameter(torch.zeros(patches))
            self.I0 = nn.Parameter(torch.zeros(patches))
            self.R0 = nn.Parameter(torch.zeros(patches))
            self.D0 = nn.Parameter(torch.zeros(patches))
            self.V0 = nn.Parameter(torch.zeros(patches))

    def get_scenario_parameters(self, scenario_id):
            # Return any scenario-specific variable needed
            raise NotImplementedError

    def forward(self) -> Tuple[torch.Tensor]:
            device = "cuda" if torch.cuda.is_available() else "cpu"
            self.to(device)

            # Initialize all disease_state variables as fractions in [0, 1]
            S = self.S0.clone().detach().requires_grad_(True)
            E = self.E0.clone().detach().requires_grad_(True)
            I = self.I0.clone().detach().requires_grad_(True)
            R = self.R0.clone().detach().requires_grad_(True)
            D = self.D0.clone().detach().requires_grad_(True)
            V = self.V0.clone().detach().requires_grad_(True)

            M = []    # cumulative deaths per week (patches × T)
            Inf = []  # cumulative infections per week (patches × T)

            beta = torch.exp(self.log_beta)

            for t in range(self.T):
                lambda_t = beta[t] * S * I
                dS_dt = -lambda_t - self.nu * S + self.omega_v * V
                dE_dt = lambda_t - self.sigma * E
                dI_dt = self.sigma * E - self.gamma * I - self.mu * I
                dR_dt = self.gamma * I - self.omega_r * R
                dD_dt = self.mu * I
                dV_dt = self.nu * S - self.omega_v * V

                S = S + dS_dt * self.h
                E = E + dE_dt * self.h
                I = I + dI_dt * self.h
                R = R + dR_dt * self.h
                D = D + dD_dt * self.h
                V = V + dV_dt * self.h

                M.append(D.clone())
                Inf.append(I.clone())
```

*(a)* 7b

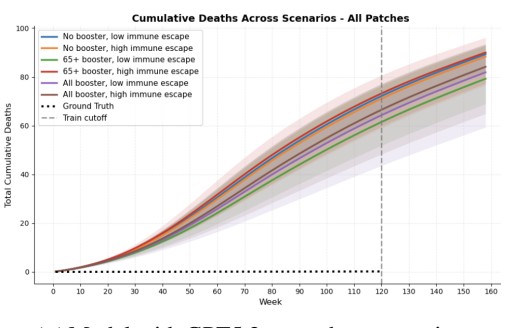

*(a)* Model with GPT5.2, zero shot prompting

```
        self.h = 40
        self.T = T + self.h
        self.patches = patches  # USA_states as patches
        self.scenario = scenario_id

        # Define the disease parameters trainable, which are shared across all scenarios. As the model will be trained on just one scenario.
        # Make disease parameters vector.
        # like there will self.T betas, self.T gammas. like ->
        self.log_beta = nn.Parameter(torch.log(torch.full((self.T,), 0.25)))
        self.sigma = nn.Parameter(torch.tensor(0.5))
        self.gamma = nn.Parameter(torch.tensor(0.1))
        self.mu = nn.Parameter(torch.tensor(0.01))
        self.omega_R = nn.Parameter(torch.tensor(0.01))
        self.omega_V = nn.Parameter(torch.tensor(0.01))
        self.nu = nn.Parameter(torch.tensor(0.01))

        # So, make sure all the shared parameters will be optimized and show good performance across all scenarios.
        # For example, if two scenarios have different infection rates, keep only one as trainable (beta), and keep a fixed variable as a multiplying factor.

        # Don't make initial conditions or scenario-specific variables trainable.

    def get_scenario_parameters(self, scenario_id):
        # Return any scenario-specific variable needed
        raise NotImplementedError

    def forward(self) -> Tuple[torch.Tensor]:
        device = "cuda"

        # Initialize all disease_state variables as fractions in [0, 1]
        S = torch.ones((self.patches, 1), device=device)
        E = torch.zeros((self.patches, 1), device=device)
        I = torch.zeros((self.patches, 1), device=device)
        R = torch.zeros((self.patches, 1), device=device)
        V = torch.zeros((self.patches, 1), device=device)
        D = torch.zeros((self.patches, 1), device=device)

        M = []    # cumulative deaths per week (patches × T)
        Inf = []  # cumulative infections per week (patches × T)

        # Retrieve scenario parameters
        # params = self.get_scenario_parameters(self.scenario)

        for t in range(self.T):
            # Implement disease_state updates using explicit dt
            # Ensure numerical stability and differentiability

            lambda_t = torch.exp(self.log_beta[t])
            dSdt = -lambda_t * S + I - self.nu * S
            dEdt = lambda_t * S * I - self.sigma * E
            dIdt = self.sigma * E - self.gamma * I - self.mu * I
            dRdt = self.gamma * I - self.omega_R * R
            dVdt = self.nu * S - self.omega_V * V
            dDdt = self.mu * I
```

*(b)* 14b

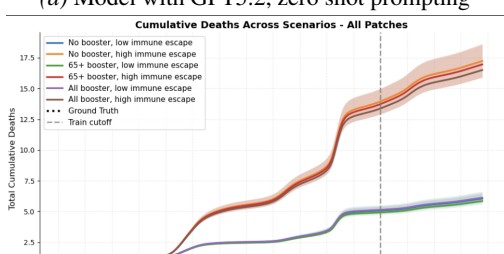

*(b)* Model with Claude Opus, zero shot prompting

*Figure 20.* Cumulative-death trajectories across scenarios from Opus 4.6, GPT-5.2, and the SMH ensemble. Opus 4.6 correctly differentiates scenarios A/B but loses separation for E/F and produces incorrect fits; GPT-5.2 shows systematically incorrect scenario ordering.

```
    def get_scenario_parameters(self, scenario_id):
        # Immune escape and vaccination schedule per scenario
        # Returns: immune_escape_rate (per week), vaccine_boost (patches, T)
        if scenario_id in ["A", "C", "E"]:
            immune_escape_annual = 0.20
        else:
            immune_escape_annual = 0.50
        immune_escape_rate = 1 - (1 - immune_escape_annual) ** (1/52)
        # Vaccine schedule: simple placeholder (no boosters except for C/E)
        vaccine_boost = torch.zeros(self.patches, self.T)
        if scenario_id in ["C", "D"]:
            # 65+ booster: assume 20% of population in each patch gets boosted at week 20 (September)
            vaccine_boost[:, 20] = 0.20
        elif scenario_id in ["E", "F"]:
            # All booster: 34% of population in each patch gets boosted at week 20
            vaccine_boost[:, 20] = 0.34
        return immune_escape_rate, vaccine_boost

    def forward(self) -> Tuple[torch.Tensor, torch.Tensor]:
        device = self.log_beta.device
        T = self.T
        N = torch.full((self.patches,), 1.0, device=device)  # normalized population
        dt = self.dt

        # Initial conditions (fractions)
        S = torch.full((self.patches,), 0.85, device=device)  # susceptible
        I = torch.full((self.patches,), 0.01, device=device)  # infectious
        R_vax = torch.full((self.patches,), 0.07, device=device)  # recovered/immune (vax)
        R_inf = torch.full((self.patches,), 0.06, device=device)  # recovered/immune (inf)
        D = torch.zeros((self.patches,), device=device)        # cumulative deaths
        C = torch.zeros((self.patches,), device=device)        # cumulative infections

        # Get scenario-specific parameters
        immune_escape_rate, vaccine_boost = self.get_scenario_parameters(self.scenario)
        vaccine_boost = vaccine_boost.to(device)
```

*(c)* 32b

*Figure 19.* Models generated by Qwen2.5-Coder variants.

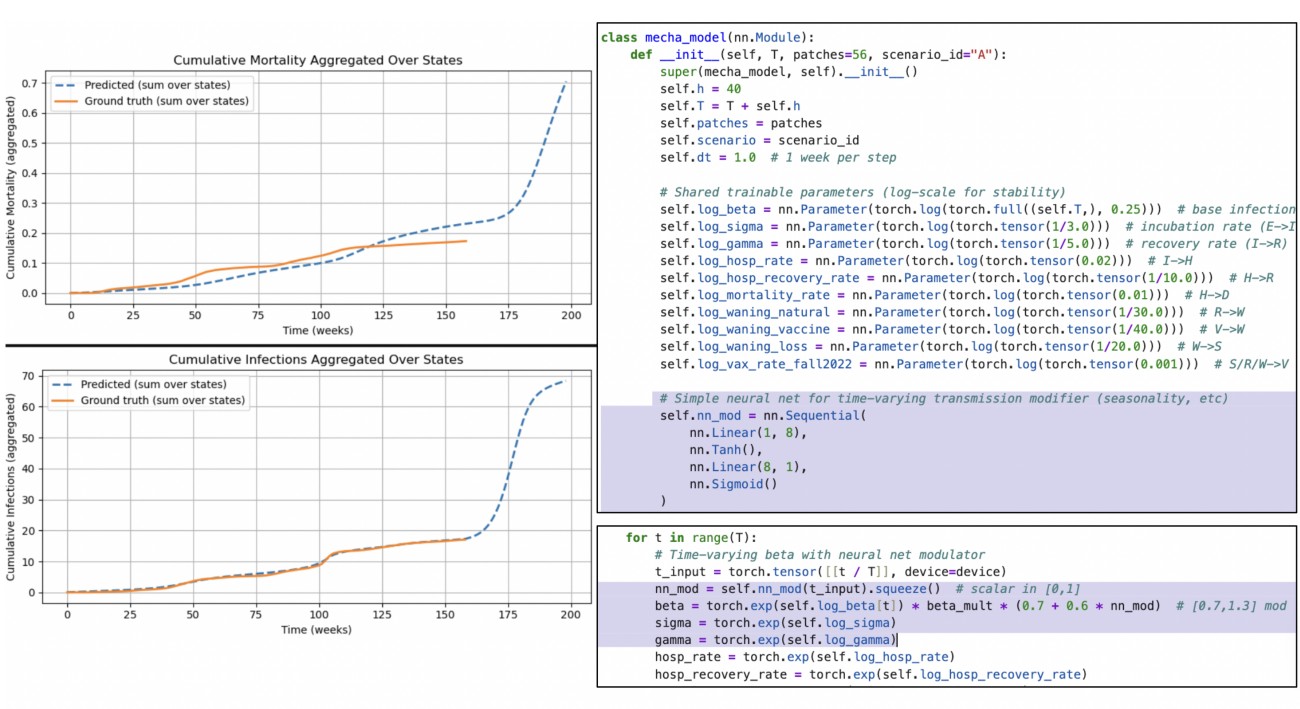

(a)Calibration using Hybrid Simulator            (b) LLM Planner generated hybrid code with time variant β

*Figure 21.* Neural Augmented Dynamics with time variant disease parameters.

