# OpenReview forum: "Agentic Framework for Epidemiological Modeling"
_ICML.cc/2026/Conference — ICML 2026 regular_

### Official Review · Reviewer_Kdz1 · 2026-03-13

**Soundness:** 2
**Presentation:** 2
**Significance:** 3
**Originality:** 3
**Overall Recommendation:** 4
**Confidence:** 4

**Summary:**

This paper introduces EPIAGENT, an LLM-based agentic framework that automates the construction of epidemiological models from natural language scenario descriptions. Traditional epidemic modeling is a slow, manual process that requires expert epidemiologists to redesign models from scratch whenever disease conditions, policies, or assumptions change. Authors tackles this problem by framing epidemic model construction as an iterative program synthesis problem. Given a natural language scenario, the system first retrieves relevant epidemiological knowledge to enrich its prompt, then generates an intermediate flow graph representing the model's compartmental structure (e.g., how populations move between Susceptible, Exposed, Infectious, Recovered states). This graph is verified for structural and epidemiological correctness before being compiled into executable PyTorch simulator code. The simulator is then calibrated to real observational data using gradient-based optimization, and a final multi-agent verification and validation stage checks that the outputs are mathematically valid, epidemiologically meaningful, and consistent across counterfactual scenarios. The key insight driving the design is that structural errors are much easier to catch at the graph level than at the code level, and that an incorrect model structure can sometimes produce superficially plausible outputs, making automated verification essential rather than optional. Experiments on COVID-19 scenario modeling case studies show that the framework produces accurate fits, correct counterfactual responses to vaccination and immune escape assumptions, and converges to valid models significantly faster than unguided LLM generation.

**Compliance With Llm Reviewing Policy:**

Affirmed.

**Final Justification:**

Most of my concerns were addressed in the rebuttal by the authors. I raised my score from 3 to 4.

**Key Questions For Authors:**

1. Do the more advance models such as Claude Opus 4.6 or GPT 5.2 still require this agentic framework or they can just accomplish it off the shelf - especially if operating under extending thinking mode.

2. In Figure 3 - what is black and gray dotted line? Can you also explain how much data was used for training and beyond what point it is forecast? and how does it performs as compared to ground truth.

3. In Figure 4 - why is (a) considered incorrect projections while (b) is considered correct projections?

**Limitations:**

Yes authors have discussed the limitations.

**Strengths And Weaknesses:**

**Strengths**

1. Paper addresses an important problem of automating the construction of epideimological models. Epidemic modeling genuinely suffers from scalability bottlenecks when policies or variants shift. The motivation for automating this workflow is well-grounded and of high significance.

2. Although AI agents for scientific discovery and modeling have been used in the past. However, to the best of my knowledge this is first work on using LLM based AI agents for epidemic modeling. The layered V&V architecture spanning structural, behavioral, and scenario-consistency checks  thoughtfully mirrors how expert epidemiologists actually validate models.  The insight that structural errors are easier to catch at the graph level than at the code level is an interesting insight.


**Weaknesees**

1. Evaluation presented in the paper seems limited.

    a.  Baselines - Table 1 compares model parameterization choices (time-invariant, time-variant, neural) but there is no comparison against human-designed models

    b. It would be helpful to see studies for other epidemics beyond Covid.

    c. The models studied also seem simple. It would be interesting to see if it can model more comple epidemic models.

    d. LLM dependcy in not explored - how will this framework perform if we use other LLM models apar from GPT-4.1.

2. Overall, as each step of verification and validation is being done using LLMs - how is soundness of the whole process is being ensured. A more robust empirical evaluations on this front are required. This might include:

    a. Some quantitative study on how many times the ouput epidemilogical model is correct or incorrect depending on the number of iterations (feedback loops) at each step of the framework depending on how complex the epidemilogical model is.

    b. What are common failure modes of the framework - where should a human verifier look for the errors in the results of the frameowrk.

3. A more broader discuss on limitations of the framework are needed (apart from the discussion in Appendix H) based on the above analysis.

---

> ### Author Rebuttal · Authors · 2026-03-31
>
> We appreciate these insights and will add a Limitations & Robustness section.
> - *a. Comparison w/ human-designed models:* We include ensemble projections from the COVID-19 Scenario Modeling Hub here for the same round, which closely align with our model’s projections (Fig 3).
> - *b. Beyond Covid:* EpiAgent can be used on FluForecasting data with only minor prompt modifications (as the task changes from projection ->forecasting). Unlike SMH, this is a forecasting task, so there are no multiple coupled scenario cases.
> https://github.com/cdcepi/FluSight-forecast-hub/tree/main
> - *c. Complexity of the epimodels:* The models formulated here incorporate age-structured vaccination and booster effects, differential immune escape, waning immunity, disease-induced deaths, and imperfect vaccination. These are not simple extensions of baseline epidemic models, rather they involve multiple interacting mechanisms that jointly shape epidemic dynamics. Accounting for these processes in a coherent framework substantially increases both the structural and analytical complexity of the model.
> - d. LLM dependency: We present results across multiple LLMs in EpiAgent, including GPT-5.2 and local models (Qwen; Reviewer3, Q2), and report the standalone model performance later.
>
> *Soundness of the framework*
> - a. Feedback loops: Feedback iterations progressively resolved failures from structural to implementation level: 17% correct generations at itr1 (scenario generation empty/crashing), 50% at itr2 (incorrect vaccination), 67%–83% at itr3–4 (flow conservation violations), and 100% by itr5
> - b. Human error checks: Common failure modes can be checked at:
>     1. Model-level checks: Verify that the compartmental structure is correct, e.g.,
>         - valid SEIRD flows
>         - all required compartments are present, transitions are epidemiologically meaningful, population is conserved.
>         - whether the model captures the scenario-specific assumptions appropriately. For example, even if vaccination is not represented through an explicit V compartment, it should still be clear that its effect is handled correctly in the model dynamics.
>     2. Projection-level checks: Verify that the outputs are consistent with epidemiological expectations and scenario definitions. For instance, projections should reflect that vaccination reduces infections and deaths, and that broader booster coverage would generally have a larger effect than booster coverage limited to high-risk groups, which in turn should outperform a no-booster setting.
>
> **Q1:Claude Opus 4.6, GPT 5.2 Performance**
> Under zero-shot prompting (no flow graph, no V\&V, no feedback), both models produced reasonable models, *but not the epidemiologically precise simulators that EpiAgent generates*.
> To ensure a fair comparison, we kept the optimization function identical across models.
> - **GPT-5.2**
>     - Generated a plausible but overly complex model
>     - Attempted RK4 but implemented incorrectly–worse than Euler
>     - Counted dS twice reinfection mass–conservation broken at ODE level(invalid transition).
>
> - **Claude Opus 4.6**
>     - Valid transitions but introduced extra states not specified in the scenario (e.g., hospitalization)
>     - Euler only but correctly implemented; didn't double count as GPT 5.2
>     - We specified the initial condition to be static, but this violates the skeleton spec–a design error not an implementation error
>     - However, the overall projections do not reflect the intended scenario differences observed in the ensemble forecasts.
>
> Projection plots: https://figshare.com/s/2c6cba536f48cb7b321e
> The figures compare cumulative-death trajectories across scenarios from Opus 4.6, GPT-5.2, and the SMH ensemble. Opus 4.6 correctly differentiates scenarios A/B but loses separation for E/F and produces incorrect fits; GPT-5.2 shows systematically incorrect scenario ordering—both diverge substantially from the ensemble reference. These results suggest that, even with the latest models, the full EpiAgent pipeline remains necessary for reliable deployment.
>
> **Q2:Figure 3**
> The dashed vertical line is the boundary between the training and projection period. This is not a forecasting task, so we deliberately do not show ground truth beyond the dashed line, as the objective is scenario projection: learning epidemiological parameter patterns from observed data (black dotted line) and projecting forward under different scenario assumptions. We use 120 weeks of data from all states for training, and then generate 40-week scenario-based projections beyond that point.
>
> **Q3:Figure 4**
> (a) is incorrect because the scenario ordering violates epidemiological expectations: no booster + high immune escape should produce the highest infections due to frequent reinfections, which is not reflected. (b) uses a different incorrect graph, yet partially recovers the correct ordering — highest burden under no booster/high immune escape, lowest under all booster/low immune escape.

---

> > ### Author Rebuttal · Reviewer_Kdz1 · 2026-04-04
> >
> > Thanks for the detailed response. Most of my concerns are addressed. I will raise my score to 4.

---

> > > ### Author Response · Authors · 2026-04-06
> > >
> > > We sincerely thank the reviewer for the positive acknowledgement. We are very pleased that our rebuttal addressed most of the concerns and are thankful that the reviewer raised the score.

---

### Official Review · Reviewer_Q2dH · 2026-03-13

**Soundness:** 2
**Presentation:** 2
**Significance:** 3
**Originality:** 3
**Overall Recommendation:** 4
**Confidence:** 3

**Summary:**

This paper presents EPIAGENT, an agentic system that automatically turns natural-language epidemic scenarios into verified mechanistic simulators. Its core innovation is an explicit flow-graph intermediate representation of disease-state transitions, which lets the system check structural correctness before generating code. Compared with naive LLM generation, EPIAGENT more reliably produces valid, interpretable epidemic models, fits observed data well, and yields epidemiologically consistent counterfactual forecasts under changing assumptions such as vaccination and immune escape level.

**Compliance With Llm Reviewing Policy:**

Affirmed.

**Key Questions For Authors:**

1. Are there any general-purpose LLM agent frameworks that could be directly compared with your proposed method?
2. It would be helpful to evaluate the framework across different LLMs, including both frontier API models and lightweight open-source models (e.g., Qwen).

**Limitations:**

yes

**Strengths And Weaknesses:**

Strengths:
The problem is well motivated. An agentic framework for accelerating and automating scenario-conditioned epidemic modeling could be highly valuable for answering what-if questions in a timely manner during the early stages of a new pandemic. The SEIR example is intuitive, the paper is easy to follow, and the experimental questions are well designed, especially the analyses of calibration error and counterfactual behavior.

Weaknesses:
My main concern is the experimental section. It seems that most of the experiments are ablation studies, with no strong baselines for comparison. This makes it harder to judge the advantage of the proposed method. In addition, some figures are not clearly explained. For example, it is unclear in Figure 3 what the black dashed line represents, what the shaded area indicates, and which curves correspond to predictions versus ground truth.

---

> ### Author Rebuttal · Authors · 2026-03-31
>
> Thank you for the helpful comment. We will add comparisons with newer models, including GPT-5.2, Opus 4.6, and the local model Qwen. We will also correct the legend in Figure~3: the black dotted line represents observed epidemiological data used for fitting; the dashed vertical line marks the projection boundary; shaded bands represent parametric uncertainty from perturbing the calibrated $\beta$ within $\pm15\%$ over 200 forward passes (10th--90th percentile interval). Ground truth is omitted in the projection period, as this is a scenario projection task.
>
> **Q1: Comparison with general-purpose LLM agent**
> Yes, we compared EpiAgent against general-purpose LLM agent frameworks as baselines, including MLGym and HDTwinGen but found them unsuitable for epidemiological modeling:
> - MLGym (Nathani et al.): Defaults to naive SEIRD models. Despite explicit prompting, it lacks the domain architecture to handle complex features like vaccination policies or immune escape.
> - HDTwinGen (Holt et al.): Produces structurally incorrect models with non-meaningful state flows that violate core mechanistic logic.
> Ultimately, these systems prioritize functional code over scientific validity and often fail to enforce the biological and physical constraints.
>
> We provide examples of naive generated models from MLGym and HDTwinGen that contain incorrect flows (e.g.,  D←fraction(I))
> https://figshare.com/s/1633a4fa9b5a04600d75
>
> **Q2: Evaluation across different LLMs**
> We thank the reviewer for this suggestion and have run EpiAgent across several models. For open-source models, we evaluated three Qwen2.5-Coder variants:
> - *7B-Instruct* failed to generate executable code, consistently violating the skeleton interface and producing code that could not run in the predefined environment.
> - *14B-Instruct* produced executable but *overly simplistic simulators* that did not reflect scenario-level differences.
> - *32B-Instruct* showed performance comparable to GPT-4.1, successfully generating scenario-consistent simulators within the EpiAgent pipeline.
>
> The screenshots of models generated by the Qwen2.5-Coder models are available here: https://figshare.com/s/932782e46a33e857f600
>
> For frontier models,
> - *GPT-5.2* paired with the flow graph and V\&V layer generated correct models.
> - However, *GPT-5.2* could not participate in the feedback loop–its content policy restricts biological experimentation-related responses, preventing actionable V\&V feedback from being returned.
>
> We also conducted zero-shot evaluations on this task using GPT-5.2 and Claude Opus 4.6 alone, without the flow-graph or V&V layers; please see Reviewer 4, Q1, for details.

---

> > ### Author Rebuttal · Reviewer_Q2dH · 2026-04-04
> >
> > Thank you for your detailed response and for carefully addressing my comments. I'll keep the positive score.

---

> > > ### Author Response · Authors · 2026-04-06
> > >
> > > We sincerely thank the reviewer for considering our rebuttal. We are glad that our response addressed the concerns, and we appreciate the reviewer for maintaining the positive score.

---

### Official Review · Reviewer_UKaR · 2026-03-13

**Soundness:** 4
**Presentation:** 4
**Significance:** 3
**Originality:** 3
**Overall Recommendation:** 5
**Confidence:** 4

**Summary:**

The paper presents EpiAgent - an agentic framework for automated epidemic modelling. The agentic pipeline consists of a few different scaffolds: 1. natural language scenarios augmented with domain knowledge, 2. epidemiological flow-graph synthesis, 3. simulation generation using an LLM planner agent, and 4. multi-agent verification and validation. The paper also provides and thorough evaluation of the full agentic pipeline with extensive ablations.

**Compliance With Llm Reviewing Policy:**

Affirmed.

**Final Justification:**

The rebuttal addressed all my raised concerns. I am keeping my recommendation to accept the paper.

**Key Questions For Authors:**

1. How would you improve the EpiAgent framework further? Would you consider adding any additional components to the framework in future work?
2. Have you tried comparing your models created with EpiAgent to mathematical DE models people create for epidemiological modelling?
3. For your consideration: it might be worth adding some extra related work in the agentic systems section:

[1] Nathani, Deepak, et al. "Mlgym: A new framework and benchmark for advancing ai research agents." arXiv preprint arXiv:2502.14499

[2] Jiang, Zhengyao, et al. "Aide: Ai-driven exploration in the space of code." arXiv preprint arXiv:2502.13138 (2025).

[3] Cui, Hao, et al. "Curie: Evaluating llms on multitask scientific long context understanding and reasoning." arXiv preprint arXiv:2503.13517 (2025).

**Limitations:**

Yes.

**Strengths And Weaknesses:**

## Strengths
* The paper is extremely well written.
    * The introduction of the epidemiology modelling using the SEIR model is clear. After reading the paper I have a much better understanding of epidemiological models.
    * The notation used in more theoretical sections is easy to follow.
    * The sections are signposted, figure 2 also includes section signposts, it all really helps with the overall paper clarity, especially given that the agentic pipeline has many components.
* The application of epidemiological modelling is very relevant (especially in the context of pandemic preparedness) and it's a great use case for agentic LLM systems.
* The results presented in the paper look quite promising.

## Weaknesses
* Just a few minor issues:
    * Results figures are not vector graphics.
    * In Equation 2, it's not explained what $P$ is. Is it $|\mathcal{X}|$?
    * Section 4.4, in the equation, what is $\hat{x}_p$?
    * In Figure 3, do you plot the observed epidemiological data? Legend is missing the dotted line (also in Figure 4).
    * Nitpick: in Figure 2, in the SEIR model could you use the same parameters as in section 3.1?

---

> ### Author Rebuttal · Authors · 2026-03-31
>
> We thank the reviewer for these helpful observations. In the revision, we will regenerate the figures as vector graphics, update Figure 2 to use SEIR parameters consistent with §3.1, and correct the legends in Figures 3 and 4.
>
> - Yes, in Equation~2 $P$ is $|\mathcal{X}|$.
> - In Section 4.4, $x_p$ and $\hat{x}_p$  denote the observed and simulated trajectories for patch $p$, and $\Delta\hat{x_p}=\hat{x}\_{p,t+1}- \hat{x}\_{p,t} $ denotes the first-order difference.
> - In Figures 3 and 4, the black dotted line represents observed data used for fitting; the right of the dashed vertical line is the projection period.
>
> **Q1: Improvement and Future Work**
> - The primary next step is full automation: given a published epidemiological paper, EpiAgent would automatically extract the scenario, construct the model, and produce comparable results without manual prompt engineering. Case Study II partially demonstrates this – EpiAgent instantiated an established behavioral SEIR model from literature, but scenario description and prompts were still manually constructed. Automating this extraction is the immediate priority.
> - A second direction is fine-tuning the LLM for epidemiological flow graph generation using curated examples from epidemiological literature, which would improve graph quality and reduce verification iterations for complex scenarios.
> - Additional directions include convergence-aware stopping criteria to address late-stage regression, spatially-structured metapopulation extensions, and Bayesian uncertainty quantification for policy use.
>
> **Q2: Comparison with DE models**
> EpiAgent does exactly this–both in Case Study I (scenario-based modeling) and Case Study II (Gozzi et al., 2025), the framework instantiates compartmental models from natural-language descriptions and calibrates them with real data. Case Study II in particular reproduces a published expert-designed DE model automatically, with fits reported in Figure 6 comparable to the original paper's results.
>
> **Q3: More Related Work**
> Thank you for the helpful pointers. We will add MLGym, AIDE, and CURIE to the related work in the agentic systems section. These works focus on recent general ML research agents and scientific reasoning benchmarks.

---

> > ### Author Rebuttal · Reviewer_UKaR · 2026-04-01
> >
> > I'm glad the authors will include the manuscript revisions they list at the beginning of their rebuttal response. My recommendation was already to accept the paper, so I will keep the same recommendation.

---

> > > ### Author Response · Authors · 2026-04-06
> > >
> > > We sincerely thank the reviewer for the positive acknowledgement and support. We are pleased that the planned revisions were well received, and we appreciate the recommendation to accept.

---

### Official Review · Reviewer_ZMPY · 2026-03-15

**Soundness:** 2
**Presentation:** 2
**Significance:** 2
**Originality:** 2
**Overall Recommendation:** 4
**Confidence:** 3

**Summary:**

The paper proposes EPIAGENT, an LLM-based framework for building epidemiological simulators from natural-language scenario descriptions. The core idea is to generate an intermediate epidemiological flow graph, verify it against domain constraints, compile it into simulator code, calibrate the simulator on data, and then use multiple agents to validate and refine the result. The paper argues that this workflow is more reliable than directly prompting an LLM to write epidemiology code, and presents case studies showing that the generated simulators can fit observed trajectories and produce plausible counterfactual projections.

**Compliance With Llm Reviewing Policy:**

Affirmed.

**Key Questions For Authors:**

What is the main novelty here beyond applying a now-standard agentic code-generation-and-verification pipeline to epidemiological simulators?

How much of the final performance comes from the flow-graph representation itself, versus the hard-coded constraints, skeleton code, and verifier feedback?

If strong scaffolding and domain constraints are essential, how autonomous is the framework in practice?

How robust is the system to imperfect or incomplete scenario descriptions? Does it fail gracefully, or does it tend to hallucinate plausible but wrong structure?

**Strengths And Weaknesses:**

Short strengths

The problem is meaningful: automating parts of epidemiological model construction could be useful in practice.
The flow-graph intermediate representation is a sensible design choice and probably the most concrete contribution in the paper.
The multi-stage verification idea is well motivated for a high-stakes domain like epidemiology.
The paper does a decent job of framing why naive code generation is not enough and why structural constraints matter.
The system is more interesting as a workflow for constrained simulator synthesis than as a new epidemiological model.

Short weaknesses

The originality is limited if framed as an epidemiology contribution. Agent-based, compartmental, and other mechanistic epidemiological models are already well established, so the paper is not novel on that axis.
The main contribution is really an automation pipeline around existing epidemiological modeling practice, not a new modeling paradigm.
The technical novelty of the agentic system feels modest: RAG, intermediate representations, code generation, verification, and feedback loops are all familiar ingredients.
A lot depends on strong scaffolding, skeleton code, explicit constraints, and the choice of underlying LLM, which makes the “autonomy” story weaker than the framing suggests.
It is not fully clear how much scientific value is being added beyond making model construction more convenient or more robust.
The evaluation seems more focused on whether the system can produce valid, plausible simulators than on whether it produces epidemiologically better or more insightful models than expert-designed alternatives.

---

> ### Author Rebuttal · Authors · 2026-03-31
>
> We thank the reviewer for raising these distinctions, as it gets at the core of our contribution.
>
> **Q1:Main novelty**
> - *Standard pipelines are insufficient.* As we discuss in the paper, we find that the standard agentic code generation and verification pipeline does not work well. A model may fit observed trajectories well while still encoding an epidemiologically incorrect structure, so low error alone is not reliable.
> - *Main novelty is the epidemiological flow graph.* A critical challenge in epidemic modeling is that a system can have a high numerical fit to observed trajectories while encoding a fundamentally incorrect causal structure—such as omitting waning immunity or suppressing reinfection. Such models are unsuitable for counterfactual reasoning or real-world policy guidance, yet these structural failures are often latent and cannot be identified through trajectory fitting alone (Fig 4b).
> - By verifying the flow graph against domain knowledge before code generation, EpiAgent ensures compartmental correctness by construction. Skipping this step yields invalid projections in >87\% of runs (§6.1). We also manually checked a subset of generated flow graphs with epidemiologists, who confirmed that they captured vaccination eligibility, waning, and immune escape correctly.
>
> **Q2:Flow-graph versus the hard-coded constraints, skeleton code, and verifier feedback**
>
> ||Correctness(%)|Main Issues|
> |-|-|-|
> |w/o hard-coded constraints|~69%|Missing depletion flows (V/R not removed on reinfection),  PyTorch errors|
> |w/o verifier feedback|~53%|Code Structurally correct (flow graph) but no recovery from mistakes without feedback|
> |w/o flow-graph|~42%|Invalid transitions (S→R direct), missing waning flows (R→S, V→S)|
>
> Here, correctness measures the fraction of generated models that are valid across scenario differentiation, compartment flow conservation, and PyTorch differentiability. Removing the flow graph caused most models to retain incorrect transitions despite verifier feedback; removing verifier feedback exposed scenario-parameter errors (especially in immune escape handling); and removing hardcoded constraints reduced overall correctness mainly due to flow-conservation violations.
>
> **Q3: Scaffolded autonomy in EpiAgent**
> We agree that EpiAgent uses fixed components like structural constraints and a hardcoded skeleton, but this is a deliberate design choice mirroring how human experts work. Rather than redefining foundational principles for each new disease, epidemiologists build on established field knowledge and tailor model structure to the conditions of the specific scenario being studied.
>
> By hardcoding theoretical foundations derived from established literature, the framework remains autonomous where it matters most: automating the selection of compartments and transitions for a given scenario. Our skeleton code and optimization setup are minimal and only fix the execution interface, not the epidemic model structure. The actual bottleneck—designing the correct compartmental structure for new variants or policy changes—is what EpiAgent automates end-to-end.
>
> The framework can generalize model formulation for epidemic scenarios beyond SARS-CoV-2 to any setting with a natural-language description and observational data. For example, applying it to flu forecasting requires only changing the disease name and target variables in the prompt. Any additional domain knowledge can be retrieved directly from epidemiological literature via RAG, ensuring the model strictly follows all structural constraints regardless of the disease or scenario.
>
> **Q4: With imperfect or incomplete scenario descriptions**
>
> When scenario descriptions are ambiguous, the system tends to generate conservative, simpler, more canonical structures rather than hallucinating plausible but wrong mechanisms.
>
> - *Counterfactual relationship inference.* Without explicit scenario-relationship guidance, EpiAgent creates independent simple simulators $f^s_{\theta_s}$ per scenario rather than a single shared-parameter model $f^s_\theta$. This confirms that while the flow graph ensures structural correctness within each scenario, inferring counterfactual relationships across scenarios $\mathcal{S}$ requires explicit specification in the prompt.
> - *Partial encoding of scenario.* The failure mode under ambiguous descriptions is under-specification (e.g., missing waning or vaccination strata) rather than structurally invalid transitions–the latter are caught and rejected at the graph level. For instance, in Case Study-I, without explicit scenario descriptions in prompt, for $\mathcal{S} = \\{A, \ldots, F\\}$, immune escape was only partially encoded: scenarios A and B shared identical compartmental structure with no $R \to S$ reinfection pathway, leaving a high immune escape structurally absent.
>
> https://figshare.com/s/0ce8d38eec0fc309847b (The attached screenshot shows that the model incorporates boosters but does not capture immune escape details.)

---

### Decision · Program_Chairs · 2026-04-30

**Decision:**

Accept (regular)

**Comment:**

This paper provides an LLM-based framework for constructing simulations of epidemiological counterfactuals. Even if many of the individual components are not highly technically novel, reviewers felt that the contribution of the overall framework in the context of epidemiology was significant, and the ability of the agent to use an intermediate representation and make well-grounded modeling decisions is a significant strength. This work has the potential to help domain experts rapidly explore different kinds of modeling assumptions/scenarios, where providing verification/supervision at the level of an intermediate flow diagram makes a lot of sense.